# High-mobility inertial domain walls driven by spin-transfer torque in a ferrimagnetic spinel oxide

Mingxing Wu [1] ✉, Shilei Ding [1], Laura van Schie[1,2], Shenghao Cai[3], Yuhao Qiu[3], Ao Du[1], Alexander E. Kossak [1], Rui Wu [4], Christian L. Degen [2], Xuegang Chen [3,5] ✉ & Pietro Gambardella [1] ✉

Efficient electrical manipulation of domain walls is key to developing magnetic devices with fast switching capabilities and low energy consumption. Here we demonstrate Bloch-type domain wall velocities exceeding 1 km s$^{-1}$ in the single-layer ferrimagnetic spinel oxide NiCo$_2$O$_4$ induced by spin-transfer torque at a current density of $2 \times 10^{11}$ A m$^{-2}$. This exceptional domain wall mobility is attributed to the combination of giant nonadiabatic spin-transfer torque, low magnetization, and high spin polarization. Additionally, we report a pronounced domain wall inertia effect in this ferrimagnet due to the large non-adiabaticity of the torque. The characteristic time for domain wall acceleration and deceleration is ~ 1 ns, shorter than that reported for typical ferromagnets. Our findings highlight the potential of spinel oxides as a promising platform for engineering high-performance domain wall devices that take advantage of ultrafast ferrimagnetic dynamics.

Current-induced domain wall (DW) motion is widely utilized to switch or modulate the magnetic configuration of magnetoelectronic devices[1–12]. Additionally, DWs can be utilized to store information[13–17], perform Boolean logic operations[18–21], and emulate neuronal functions[22–24] in narrow magnetic strips, so-called DW racetracks. In such systems, DW motion can be driven by spin-transfer torque (STT), arising from an electric current flowing directly through the magnetic layer[25–27], by spin-orbit torque (SOT), typically generated by a current in an adjacent heavy metal layer[28–32], and by magnon torque, originating from either coherent spin-wave[33–35] or incoherent thermal-magnons[36,37]. SOT has long been widely regarded as the more effective and versatile mechanism for DW manipulation compared to STT[38,39]. For instance, an ultrafast velocity of 5.7 km$^{-1}$ has been reported in Pt/CoGd under a current density of $4.2 \times 10^{11}$ A m$^{-2}$ and an in-plane magnetic field of 140 mT at angular momentum compensation temperature[40]. Nevertheless, STT offers several complementary features relative to SOT: it does not require an in-plane bias magnetic field for DW motion; it drives DW in a single-layer magnetic structure, eliminating the need for a heavy-metal spin source; it is effective for both Néel- and Bloch-type DWs; and the direction of DW motion can be easily controlled by the polarity of the applied current, irrespective of DW chirality. Notably, recent studies suggest that STT can achieve comparable performances to SOT[41–43], renewing interest in STT-based devices. This motivates the search for material properties and systems that lead to large STT-driven DW mobility (defined as velocity per current density) and low energy consumption, the two crucial parameters determining the performance of DW devices.

The STT-induced DW velocity at steady state is $\nu = \frac{\beta}{\alpha} u$ below the Walker breakdown, where $\beta$ is the nonadiabatic torque parameter, $\alpha$ the magnetic damping, and $u = \frac{g \mu_B P}{2 e M_s} j$ the spin-drift velocity with

[1]Department of Materials, ETH Zurich, Zurich, Switzerland. [2]Department of Physics, ETH Zurich, Zurich, Switzerland. [3]Center of Free Electron Laser & High Magnetic Field, and Leibniz International Joint Research Center of Materials Sciences of Anhui Province, Anhui University, Hefei, China. [4]Spin-X Institute, School of Physics and Optoelectronics, State Key Laboratory of Luminescent Materials and Devices, Guangdong-Hong Kong-Macao Joint Laboratory of Optoelectronic and Magnetic Functional Materials, South China University of Technology, Guangzhou, China. [5]State Key Laboratory of Opto-Electronic Information Acquisition and Protection Technology, Anhui Provincial Key Laboratory of Magnetic Functional Materials and Devices, Anhui University, Hefei, China. ✉e-mail: mingxing.wu@mat.ethz.ch; xgchen@ahu.edu.cn; pietro.gambardella@mat.ethz.ch

current density $j$, $P$ the spin polarization, $M_s$ the saturation magnetization, $\mu_B$ the Bohr magneton, $g$ the Landé factor, and $e$ the electron charge[25–27]. A larger nonadiabatic torque simultaneously enhances the DW mobility and eases depinning, thereby reducing the operating current density[44]. Meanwhile, low magnetization with high spin polarization enables efficient magnetization rotation, thereby promoting DW motion. In light of these considerations, ferrimagnets stand out as attractive materials for DW devices owing to their characteristic low magnetization and significant nonadiabatic torques[45,46].

To date, several ferrimagnetic materials have been reported to exhibit relatively high performance in STT-driven DW devices. For example, current-driven DW motion was studied in the ferrimagnetic Heusler alloys $X_3Z$ (X = Mn, Z = Ge, Sn, Sb)[47]. A DW velocity exceeding 100 m s$^{-1}$ was obtained with a current density in the order of 10$^{11}$ A m$^{-2}$. Moreover, a high DW velocity was observed in Mn$_4$N and Mn$_{4-x}$Ni$_x$N thin films with inverted perovskite structure[42,43], where DWs can be accelerated up to 2 km s$^{-1}$. However, achieving such a high velocity requires a current density of $1 \times 10^{12}$ A m$^{-2}$. Achieving high DW velocity at low current density thus remains a central objective in the development of energy-efficient STT devices. Spinel oxides, with the general chemical formula $AB_2O_4$, exhibit diverse spin and electronic properties, offering new opportunities for spintronic applications[48,49]. In particular, the ferrimagnetic compound $NiCo_2O_4$ (NCO)[50,51], with high spin polarization, high electrical conductivity, and low magnetization, holds great promise for high-velocity, low-power DW racetrack devices. Nevertheless, current-driven DW motion remains unexplored in this class of materials.

In this study, we demonstrate efficient current-driven DW motion in a perpendicularly magnetized NCO film combining high DW mobility and low energy dissipation. These features are maintained over a wide range of film thicknesses. To investigate the underlying DW characteristics, we employed scanning nitrogen-vacancy (NV) magnetometry to resolve the DW structure and chirality. Measurements of DW dynamics were carried out using magneto-optical Kerr effect (MOKE) microscopy. We report a DW velocity exceeding 1 km s$^{-1}$ with a current density of only $2 \times 10^{11}$ A m$^{-2}$, corresponding to a DW mobility of $5.7 \times 10^{-9}$ m$^3$ A$^{-1}$ s$^{-1}$. This excellent performance is attributed to the unique intrinsic properties of NCO, including a giant nonadiabatic STT, low magnetization, and high spin polarization. An in-plane magnetic field applied parallel to the wall magnetization produces a moderate decrease of the DW velocity, independently of field or current polarity, consistent with an increase of the DW width and decrease of nonadiabaticity. In contrast, a field applied parallel to the current produces no measurable change of DW velocity, consistently with Bloch DW dynamics below the Walker limit. Furthermore, we observe a pronounced DW inertia effect, also ascribed to the large nonadiabaticity. This effect enables the extraction of key STT parameters, such as the characteristic time for DW acceleration and deceleration, as well as the adiabatic and nonadiabatic torque coefficients. Our study provides guidelines for developing efficient STT-based racetrack memory devices using ferrimagnetic spinel oxides.

## Results
### Epitaxial film growth and characterization
Epitaxial NCO films with thicknesses from 10 to 30 unit cells (uc) (1 uc = 0.8 nm), grown on (001)-oriented MgAl$_2$O$_4$ (MAO) substrates, were used for the DW racetrack devices (Methods). The NCO exhibits an inverse spinel structure, as shown in Fig. 1a. In each unit cell, the

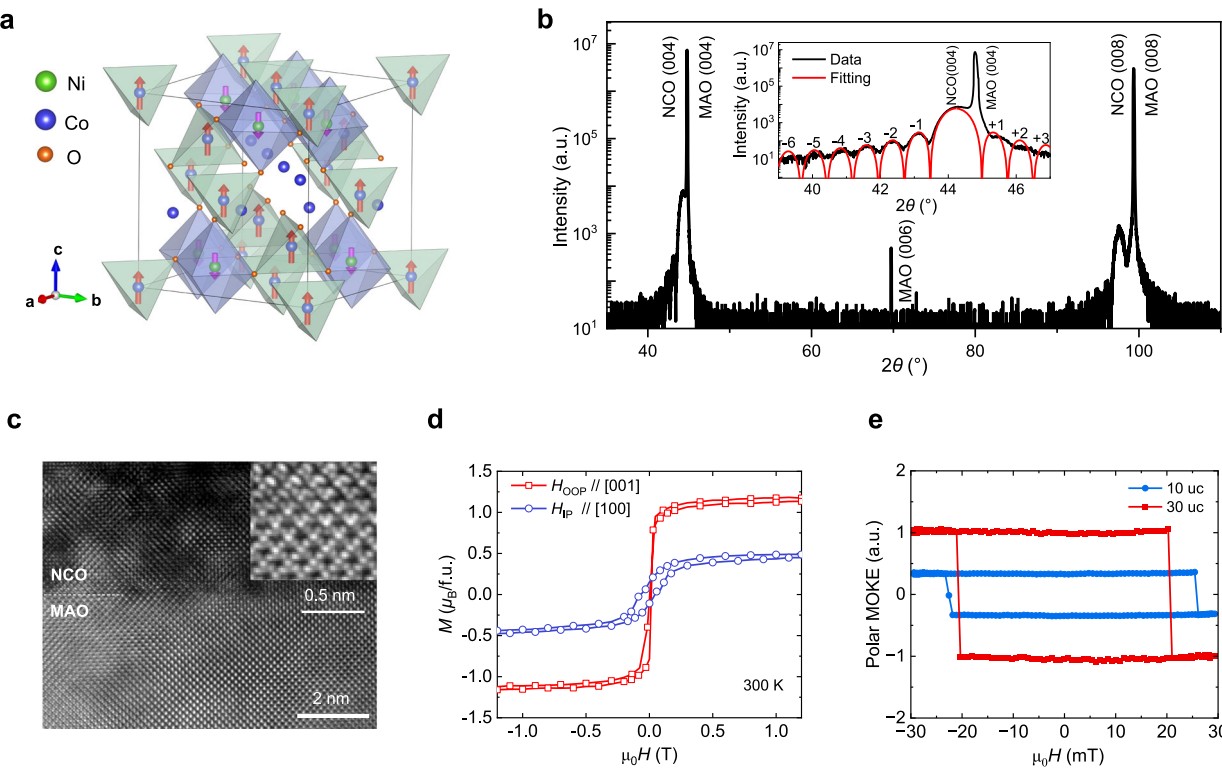

**Fig. 1 | Characterizations of NCO epitaxial films. a** Schematic spin structure of ferrimagnetic NCO consisting of antiferromagnetically coupled Ni and Co spins on octahedral and tetrahedral sites, respectively. The structure was generated using VESTA software[91]. **b** $\theta$–$2\theta$ XRD scan of a 16 uc NCO film. Inset: expanded $\theta$–$2\theta$ scan near the (004) diffraction peak with the fit of the Laue oscillations (red line). **c** STEM image of a 10 uc NCO film. Inset: high-resolution STEM image showing the NCO crystal structure. **d** In-plane (IP) and out-of-plane (OOP) *M-H* hysteresis loops of a 12 uc NCO film measured by SQUID magnetometry. **e** Polar MOKE hysteresis loops of 10 and 30 uc NCO films. The PMA is stabilized within the investigated thickness range from 10 to 30 uc.

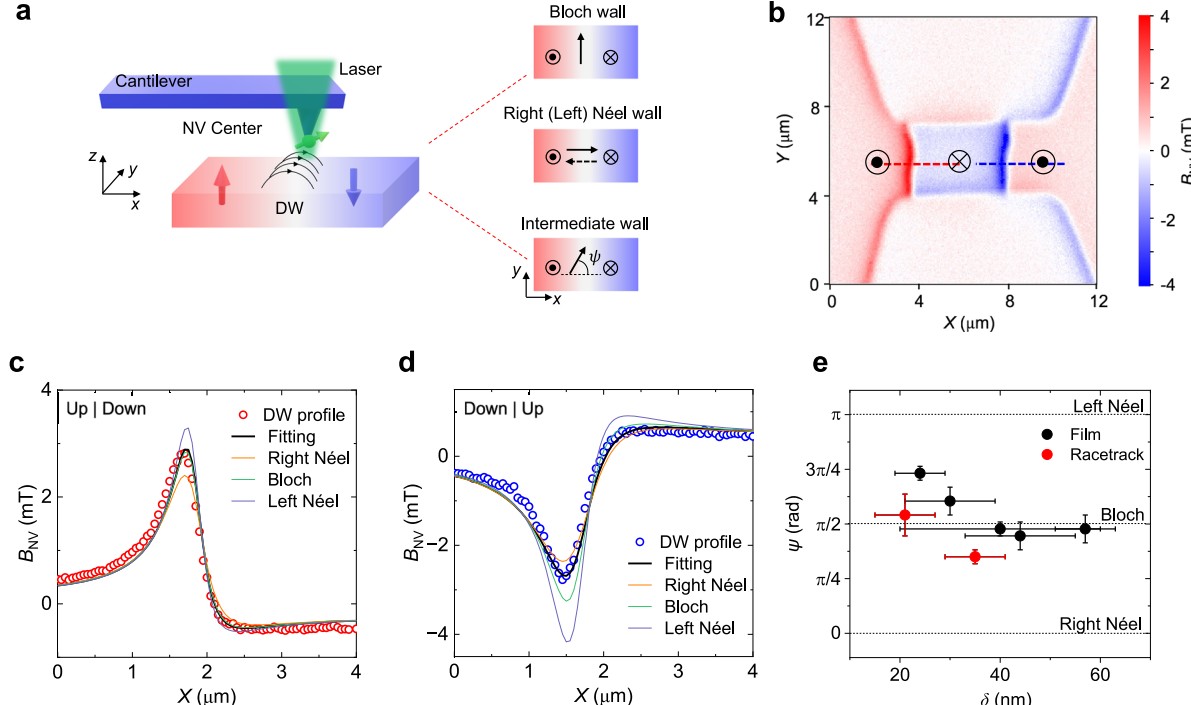

**Fig. 2 | DW structure and chirality measured by scanning NV magnetometry.**
**a** Schematic of a scanning NV microscopy sensing the stray field of a DW. The DW chirality is described by the angle $\psi$ between the wall magnetization (black arrows) and the direction perpendicular to the wall ($x$-direction). The wall configurations (top view) are depicted for a Bloch wall ($\psi = \frac{\pi}{2}$), a right or left Néel wall ($\psi = 0$ or $\pi$), and an intermediate wall with arbitrary $\psi$, respectively. **b** Map of the magnetic stray field projected on the NV spin axis ($B_{NV}$) in a DW racetrack. Two walls with Up|Down and Down|Up configurations are created by applying a

magnetic field. The red and blue dashed lines indicate the line scans reported in panels c and d, respectively. **c, d** Stray field profiles of Up|Down and Down|Up DWs in along with their corresponding fits. The calculated line profiles assuming a pure Bloch wall (green line), a right Néel wall (yellow line), and a left Néel wall (cyan line) are also plotted for comparison. **e** Distribution of chirality and DW width for both film and racetrack regions. The error bars are the standard deviations of the fits.

magnetic moments are carried by 8 Ni ions on octahedral sites and 8 Co ions on tetrahedral sites, which are antiferromagnetically aligned. The resulting uncompensated magnetic moments give rise to ferrimagnetism. The Curie temperature of NCO is approximately 420 K, well above room temperature[50]. Moreover, NCO is a half-metal with a resistivity of ~ 810 μΩcm. The minority-spin subbands from Ni are dominant at the Fermi surface, resulting in a strong negative spin polarization ($P = -0.73$)[52]. The X-ray diffraction (XRD) spectrum in Fig. 1b demonstrates the epitaxial growth of the NCO film with a (001) orientation. The high crystalline quality is further evidenced by the presence of Laue oscillations in the expanded diffraction pattern around the (004) peak (inset). The high-resolution scanning transmission electron microscopy (STEM) image in Fig. 1c reveals the well-ordered interface between NCO film and MAO substrate, confirming the high-quality epitaxial growth of the inverse spinel structure. The magnetic properties were characterized using superconducting quantum interference device (SQUID) magnetometry. The in-plane and out-of-plane magnetic hysteresis loops measured at 300 K are presented in Fig. 1d. The out-of-plane $M$-$H$ hysteresis loop displays a sharp switching behavior, with a saturation magnetization of 1.1 $\mu_B$ per formula unit (150 kA/m), consistent with a previous report[50]. In contrast, the in-plane $M$–$H$ loop does not reach saturation even at magnetic fields exceeding 1 T, indicating strong perpendicular magnetic anisotropy (PMA) of the NCO film. To further characterize the PMA, we measured $M$-$H$ loops using polar MOKE, as shown in Fig. 1e. We find that strong PMA is sustained across thicknesses from 10 to 30 uc, with comparable coercivities of about 20 mT. This makes NCO highly favorable for DW racetrack memory applications.

## DW structure and chirality

Before investigating the current-driven DW dynamics, we examined the DW structure and chirality in a 30 uc NCO using scanning NV magnetometry (Methods)[53–55]. Near the DW, the spatially nonuniform magnetization has the following projection onto the coordinate system[53–55]:

$$M_x(x) = M_s \frac{\cos\psi}{\cosh(x/\delta)},$$
$$M_y(x) = M_s \frac{\sin\psi}{\cosh(x/\delta)}, \qquad (1)$$
$$M_z(x) = -M_s \tanh(\frac{x}{\delta}),$$

Where $x$ is the direction perpendicular to the wall, $\delta$ is the DW width, and the angle $\psi$ denotes the DW chirality. $\psi = \frac{\pi}{2}$ corresponds to a Bloch wall, whereas $\psi = 0$ or $\pi$ corresponds to a right or left Néel wall, respectively (Fig. 2a). The magnetization-induced stray field varies in space depending on DW chirality, and its spatial profile can be probed using a diamond tip containing a single NV center. Figure 2b shows a map of the magnetic stray field over the 30 uc NCO racetrack, with an Up|Down and a Down|Up wall generated by an appropriate magnetic field. To extract the DW width and chirality, we fitted the stray-field wall profiles, as shown in Fig. 2c, d. We obtained $\delta = 21 \pm 6$ nm and $\psi = 1.7 \pm 0.3$ rad for the Up|Down wall in Fig. 2c. Similarly, $\delta = 35 \pm 6$ nm and $\psi = 1.1 \pm 0.1$ rad was determined for the Down|Up wall in Fig. 2d. Although $\psi$ appears to be close to $\frac{\pi}{2}$, the uncertainty is too large to conclusively determine the DW chirality. Therefore, we scanned a larger region of the NCO film to obtain an extended data set on DWs with different orientations (Supplementary Note 1). The distribution of

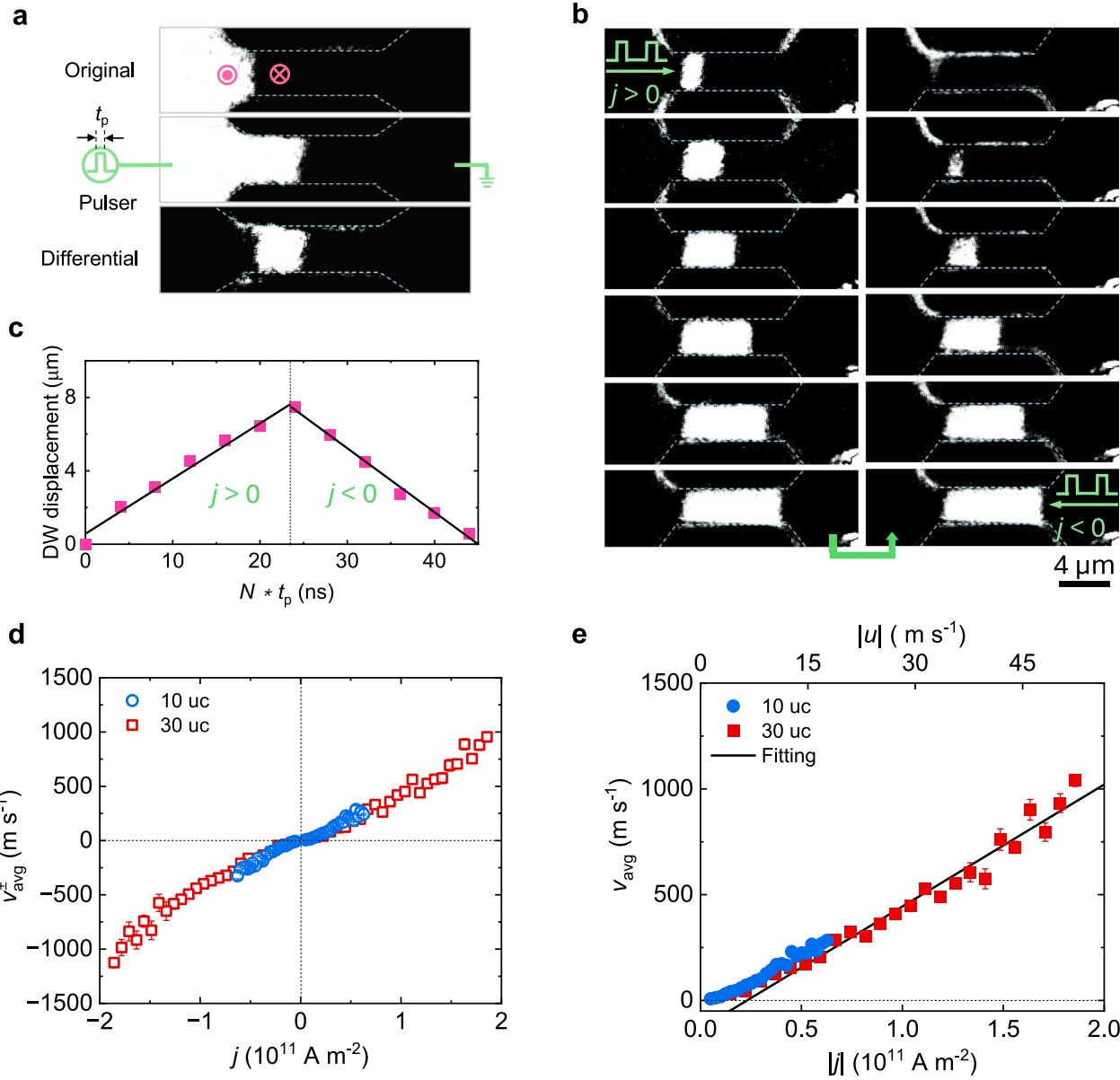

**Fig. 3 | Current-driven DW motion and velocity. a** Illustration of differential MOKE imaging for measuring DW displacements. The differential image (bottom) is obtained by subtracting a reference image (top) from the final image (middle) after application of a current pulse of duration $t_p$. **b** Sequence of DW displacements under a series of positive and subsequent negative pulses. The DW propagates in the same direction as the applied current. The current density and $t_p$ are set to $8.1 \times 10^{10}$ A m$^{-2}$ and 2 ns, respectively. **c** DW displacements after $N$ current pulses. The DW velocity is obtained as the slope of the linear fit of the displacements. **d** DW velocity as a function of current density for 10 and 30 uc NCO for positive ($v_{avg}^+$) and negative ($v_{avg}^-$) current pulses. **e** Average DW velocity induced by STT as a function of current density $|j|$ and spin-drift velocity $|u|$. The black line is the linear fit from which the DW mobility is extracted. The error bars are the standard deviations of the fits.

chirality with respect to the DW width, as shown in Fig. 2e, indicates a Bloch-type DW configuration for NCO. In films with PMA, the magnetostatic energy inhibits the formation of Néel walls[56,57], unless additional energy terms are introduced, for example, via the Dzyaloshinskii–Moriya interaction (DMI)[54]. The Bloch-wall chirality suggests negligible DMI in NCO. This is also confirmed by the in-plane magnetic field dependence of the DW velocity (Supplementary Note 2).

**Efficient current-driven DW motion**

The DW velocities in the racetrack were measured by differential MOKE imaging (Methods), as shown in Fig. 3a. After DW nucleation by an external field, a current pulse of duration $t_p$ was injected in the racetrack and the resulting DW displacement was captured by MOKE.

Here, we define a positive current flowing from left to right of the images and a positive DW displacement as a left-to-right motion. To obtain an accurate measurement of the DW velocity, a series of pulses was applied to drive the DW displacements. An exemplary sequence of DW motion is shown in Fig. 3b. We find that the DWs move in the same direction as the injected current, opposite to the direction of electron flow. This behavior is consistent with the negative spin polarization of NCO derived from theoretical calculations[58,59] and magnetic tunnel junction measurements[52] (see Supplementary Note 3 for more details).

The average DW velocity $v_{avg}^\pm$ for positive (+) and negative (-) current is extracted from linear fittings of the displacements in the measured sequence, as shown in Fig. 3c. The method used for determining the DW position is explained in Supplementary Note 4. Figure 3d presents the DW velocity as a function of current density. The

DW velocities are identical for both Up|Down and Down|Up configurations. Moreover, they are symmetrically centered with respect to in-plane magnetic bias fields $H_x$ and $H_y$ under both positive and negative current polarities (Supplementary Note 2). This is consistent with the absence of SOT in NCO, indicating STT as the sole current-driven mechanism for DW motion. Moreover, the DW velocity is similar in 10 and 30 uc racetracks (Fig. 3d). Such thickness-independent behavior is consistent with the volumetric character of STT-induced DW motion, further excluding interfacial effects as a cause of DW motion in NCO. The STT-induced DW velocity has odd parity with respect to the current direction, whereas other effects, such as Joule heating (Supplementary Note 5) and thickness nonuniformity along the racetrack, are typically independent of the current direction. Thus, the odd STT contribution to DW motion can be extracted using $v_{\mathrm{avg}} = (v_{\mathrm{avg}}^+ - v_{\mathrm{avg}}^-)/2$ and is shown in Fig. 3e. The comparison between Fig. 3d and e shows that STT dominates the DW dynamics. Remarkably, we observe a finite $v_{\mathrm{avg}} \sim 7\,\mathrm{m\,s^{-1}}$ with a current density of only $5 \times 10^9\,\mathrm{A\,m^{-2}}$. The current density required to depin the DW is one to two orders of magnitude smaller than those reported for ferromagnetic[60–63], ferrimagnetic[42,43,47,64–67], and synthetic antiferromagnetic[16,68,69] thin films, and is comparable to that observed in bulk single-crystal antiferromagnetic racetracks[70,71]. Such a low current density is attributed to the large nonadiabaticity and the high crystalline quality of the NCO film, which minimizes the intrinsic and extrinsic pinning, respectively[72]. Moreover, we observe a DW velocity exceeding $1\,\mathrm{km\,s^{-1}}$, with a current density of only $2 \times 10^{11}\,\mathrm{A\,m^{-2}}$ (see Supplementary Note 6 for a discussion on the theoretical velocity limit). According to the one-dimensional model[25–27], $v_{\mathrm{avg}} = \frac{\beta}{\alpha}u$ below the Walker breakdown and $v_{\mathrm{avg}} \approx u$ above it. In ferromagnets, $\beta$ is typically smaller than $\alpha$[26], and the STT efficiency is therefore limited by $u$, which represents the rate of spin angular momentum transfer from the electric current to the DW magnetization. However, in NCO, the observed DW velocity is more than one order of magnitude larger than the spin-drift velocity (i.e., $v_{\mathrm{avg}} \gg |u|$), as shown in Fig. 3e (see Supplementary Note 7 for DW dynamics formulation for NCO). This is possible only in the absence of Walker breakdown and if the nonadiabaticity satisfies $\frac{\beta}{\alpha} \gg 1$[61]. Under these circumstances, the nonadiabaticity can be extracted from the linear fit of Fig. 3e, yielding $\frac{\beta}{\alpha} \approx 20$. A further evaluation of the $\alpha$ and $\beta$ coefficients is presented in the Discussion Section.

## DW inertia

The spin-polarized current interacting with a DW induces two primary effects on its dynamics. The adiabatic STT causes a rotation of the magnetization within the wall plane, leading to translational motion. Simultaneously, the nonadiabatic STT tilts the magnetization out of the wall plane, leading to the deformation of the DW structure[44]. The magnetization tends to relax back to its equilibrium state after the current is turned off. This relaxation process gives rise to a DW inertial effect, manifesting as continued DW motion even after the termination of the current pulse[73,74]. The inertia effect is more pronounced in magnetic systems with large nonadiabaticity[75], and can be described using the one-dimensional DW model (Supplementary Note 8)[27,73–76]. The analytical expression of the instantaneous DW velocity $v(t)$ under a current pulse of duration $t_p$ reads[76]:

$$v(t) = \begin{cases} \frac{\beta}{\alpha}u\left(1 - e^{-\frac{t}{\tau}}\right) & \text{for } 0 \le t < t_p \\ \frac{\beta}{\alpha}u\left(1 - e^{-\frac{t_p}{\tau}}\right)e^{-\frac{t-t_p}{\tau}} & \text{for } t \ge t_p \end{cases} \quad (2)$$

Here $\tau = \frac{1+\alpha^2}{\alpha\gamma H_K}$ is a characteristic time for DW acceleration and deceleration, $H_K$ is the transverse anisotropic field which induces a preferred Bloch-wall chirality, $\gamma$ is the gyromagnetic ratio. According to Eq. (2), when $t_p \gg \tau$, the DW is accelerated up to a terminal velocity $\frac{\beta}{\alpha}u$, which is determined by the nonadiabatic STT. In this regime, the DW predominantly moves at the terminal velocity, and the inertial contribution becomes negligible. On the other hand, when $t_p \sim \tau$, the DW does not have sufficient time to reach the terminal velocity and starts to decelerate once the pulse is turned off. Figure 4a illustrates this situation using exemplary parameters of $t_p = 1\,\mathrm{ns}$ and $3\,\mathrm{ns}$, $\tau = 1\,\mathrm{ns}$, and $\frac{\beta}{\alpha}\mu = 300\,\mathrm{m\,s^{-1}}$. Clearly, the DW travels a longer distance during deceleration than acceleration when $t_p \sim \tau$. In the experiment, the DW velocity is measured via the DW displacement, and can be expressed as $v_{\mathrm{avg}} = \frac{1}{t_p}\int_0^\infty v(t)dt$. Consequently, a shorter $t_p$ results in a higher $v_{\mathrm{avg}}$. This phenomenon is evidenced by experimental results presented in Fig. 4b. The DW displacement under pulse injection exhibits a steeper slope for $t_p = 1\,\mathrm{ns}$ compared to $t_p = 3\,\mathrm{ns}$, indicating a higher measured velocity for a shorter pulse. To further investigate the inertia effect in NCO, we measured $v_{\mathrm{avg}}$ as a function of $t_p$ under various pulse current densities, as shown in Fig. 4c. A significant increase in $v_{\mathrm{avg}}$ is observed as $t_p$ decreases. To fit these curves, we compute $v_{\mathrm{avg}}$ from Eq. (2):

$$v_{\mathrm{avg}} = \frac{\beta}{\alpha}u\left(1 + \frac{\tau}{t_p}e^{-\frac{t_p}{\tau}}\right) \quad (3)$$

From the fitting of $v_{\mathrm{avg}}$ with respect to $t_p$, two key parameters can be extracted: $\frac{\beta}{\alpha}u$ and $\tau$. These values are plotted as a function of $|u|$ in Fig. 4d. The linear dependence of $\frac{\beta}{\alpha}u$ on $|u|$ yields the slope $\frac{\beta}{\alpha} = 17.0 \pm 0.3$, which is comparable to the estimate of $\frac{\beta}{\alpha}$ obtained from Fig. 3e. On the other hand, $\tau$ shows a weak dependence on $|u|$, with a magnitude of $0.9 \pm 0.2\,\mathrm{ns}$.

## Discussion

The very high DW mobility of NCO observed in our experiments indicates that ferrimagnetic spinel oxides are optimally suited for efficient STT-driven DW motion, thanks to the combination of large nonadiabaticity and high spin polarization. The DW inertia further enables us to extract the characteristic time for DW acceleration and deceleration in this ferrimagnetic NCO. The extracted $\tau \lesssim 1\,\mathrm{ns}$ is notably shorter than that observed in ferromagnets, such as for STT-driven DW in permalloy (11.5 ns)[75], and SOT-driven DW in W/Co$_{20}$Fe$_{60}$B$_{20}$ (2-5.5 ns)[76]. The short characteristic time suggests a small effective DW mass and a large DW acceleration (inversely proportional to $\tau$) in NCO, thereby facilitating ultrafast DW dynamics in the ferrimagnetic spinel oxide. Moreover, the effective parameters $\alpha$ and $\beta$ that determine the DW velocity can be separately calculated from the combination of characteristic time $\tau$ and nonadiabaticity $\frac{\beta}{\alpha}$. We estimate $\alpha = 0.08 \pm 0.03$ by taking $\tau = 0.9\,\mathrm{ns}$, $\gamma = 17.6\,\mathrm{MHz/Oe}$, and $H_K = \frac{\ln(2)}{\delta}tM_s = 800 \pm 200\,\mathrm{Oe}$, where $t$ is the film thickness[30,77], $\delta = 39 \pm 13\,\mathrm{nm}$, and $M_s = 150\,\mathrm{kA/m}$ for 30 uc NCO. From $\frac{\beta}{\alpha} = 17$ we then obtain $\beta = 1.4 \pm 0.5$ for NCO. The magnitude of $\beta$ is about three times larger than reported for ferrimagnetic GdFeCo[45], where $\beta = -0.5$. This difference likely arises from intrinsic material parameters and may also be influenced by the experimental approaches used to extract $\beta$—namely, from DW velocity and inertia measurements in NCO, versus DW mobility fitting across the compensation point in GdFeCo[27,73–76].

The nonadiabatic torque arises from the misalignment of electron spins as they traverse a magnetic texture. It is generally small in ferromagnets but can become significant in systems with staggered antiferromagnetic order. Phenomenologically, the nonadiabatic parameter can be expressed as $\beta = \left(\frac{\lambda_J}{\lambda_{sf}}\right)^2$, where $\lambda_J = \sqrt{\frac{2\hbar D_0}{J}}$ and $\lambda_{sf} = \sqrt{2D_0\tau_{sf}}$ are the s-d exchange length and spin-flip length, $D_0$, $J$, and $\tau_{sf}$ denote spin diffusion constant, s-d exchange interaction energy, and spin-flip relaxation time, respectively[27]. The s-d exchange originates from the interaction between itinerant s electrons and localized d electrons, and $\lambda_J$ is thus independent of the type of sublattices coupling. In contrast, $\lambda_{sf}$ depends strongly on the sublattice coupling, as spin-flip processes are highly sensitive to the magnetic

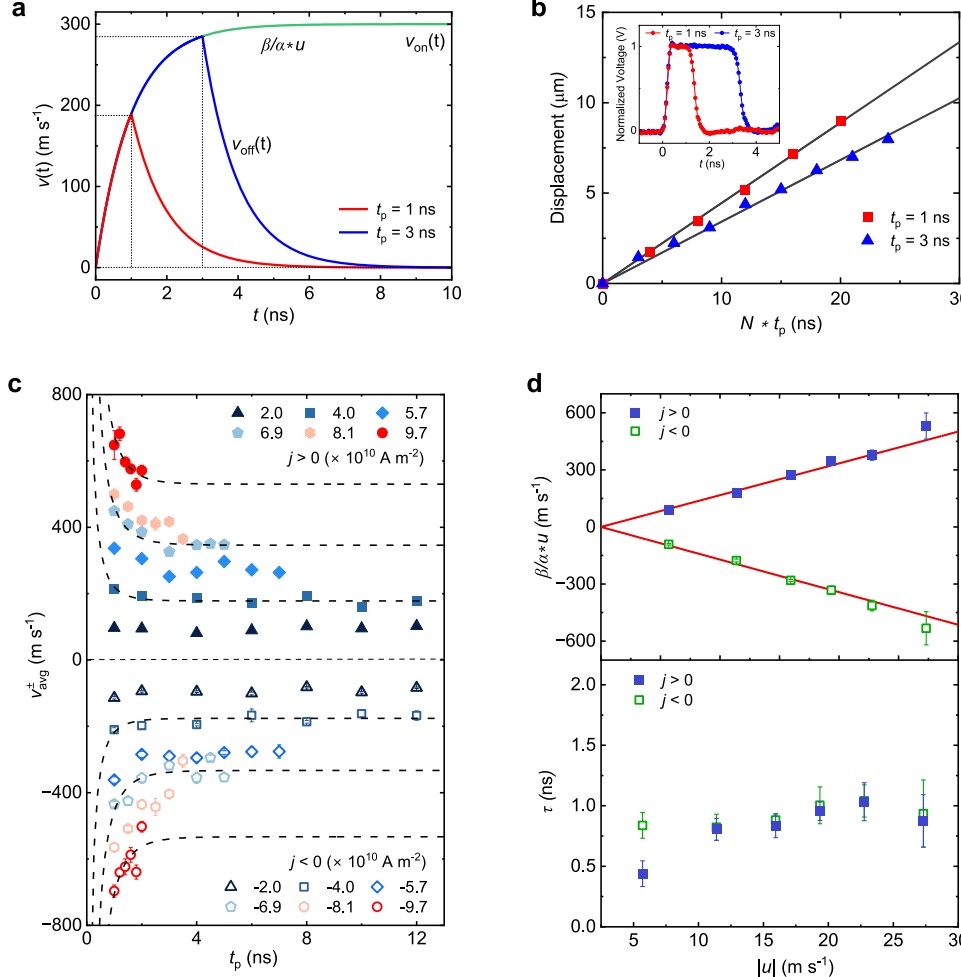

**Fig. 4 | DW inertia due to large nonadiabatic STT. a** Analytical calculations of instantaneous DW velocity $v(t)$ for pulses of duration $t_p = 1$ (red line), 3 ns (blue line), and > 10 ns (green line). Calculation parameters: $\frac{\beta}{\alpha}u = 300$ m s$^{-1}$, $\tau = 1$ ns. **b** Experimental DW displacements due to repeated pulsing with $t_p = 1$ ns and 3 ns. The black lines are linear fits to the data. The inset shows the pulse shapes of the 1-ns- and 3-ns-long pulses measured by an oscilloscope after transmission through the sample (see Supplementary Note 9 for more details). **c** DW velocity $v_{avg}^{\pm}$ as a function of $t_p$ under different pulse current densities with positive and negative polarities. The dashed lines are exemplary fits using Eq. (3) for the current densities of $\pm 4.0$, $\pm 6.9$, and $\pm 9.7 \times 10^{10}$ A m$^{-2}$. **d** Values of $\frac{\beta}{\alpha}u$ and $\tau$ extracted from the fits as a function of $|u|$. The error bars are the standard deviations of the fits.

structure. Recent theoretical calculations revealed a strongly reduced $\tau_{sf}$ when transitioning from a ferromagnetic to an antiferromagnetic sublattice[78]. In a ferromagnet, $\lambda_{sf}$ is generally on the order of several nanometers[79], whereas in an antiferromagnet it can be as short as 0.5 nm[80]. Assuming values of $\lambda_J = 1$ nm[27], $\lambda_{sf} = 5$ nm[27] for a typical ferromagnet and $\lambda_{sf} = 0.5$ nm[80] for an antiferromagnet, we estimate $\beta = 0.04$ and 4, respectively. The small value of $\beta$ in ferromagnets is in agreement with previous reports, for example, $\beta \approx 0.1$ for permalloy[81,82] and $\beta \approx 0.02$ for CoNi[83]. In contrast, $\beta$ can be up to two orders of magnitude larger in an antiferromagnetically coupled sublattice owing to spin mistracking.

Measurements of $v_{avg}$ as a function of in-plane magnetic bias fields $H_x$ and $H_y$ for both positive and negative currents reveal that the DW motion is insensitive to $H_x$ while it decreases linearly with $|H_y|$ (Supplementary Note 2). This behavior is consistent with STT-driven DW dynamics. Moreover, as the DW width is expected to increase proportionally to $H_y$ for small field/uniaxial anisotropy field ratios, the decrease of $v_{avg}$ may be attributed to a decrease of $\beta$, which is predicted to scale inversely with DW width as spin mistracking requires large magnetization gradients[84–86].

A comparison between DW mobility and displacive energy consumption provides further insight into the performance of NCO and DW devices across different material classes. Figure 5a presents the DW mobility with respect to the saturation magnetization ($M_s$) for various ferromagnetic, ferrimagnetic, and antiferromagnetic STT racetrack devices. The raw data of velocity versus current density for the extraction of the mobility is reported in Supplementary Note 10. In most materials, the DW mobility scales inversely with $M_s$, consistent with predictions from the one-dimensional model. However, the DW mobility in NCO is exceptionally high despite its finite $M_s$, deviating significantly from the above trend. This enhanced DW mobility is attributed to the giant nonadiabatic STT and high spin polarization of NCO. Further improvements in DW mobility may be obtained by tuning the magnetization and angular momentum towards the respective compensation points[43,45,87].

For assessing the energy consumption of DW racetrack devices, a practical metric is the volume-normalized energy consumption required to displace a DW $\xi_V = \frac{j^2 \rho}{v_{avg}}$ (see Supplementary Note 11 for more details)[88]. Here $j^2 \rho$ is the power density under a current density $j$ injected into a material with resistivity $\rho$. This metric enables the direct comparison of the energy required to displace a DW per unit length

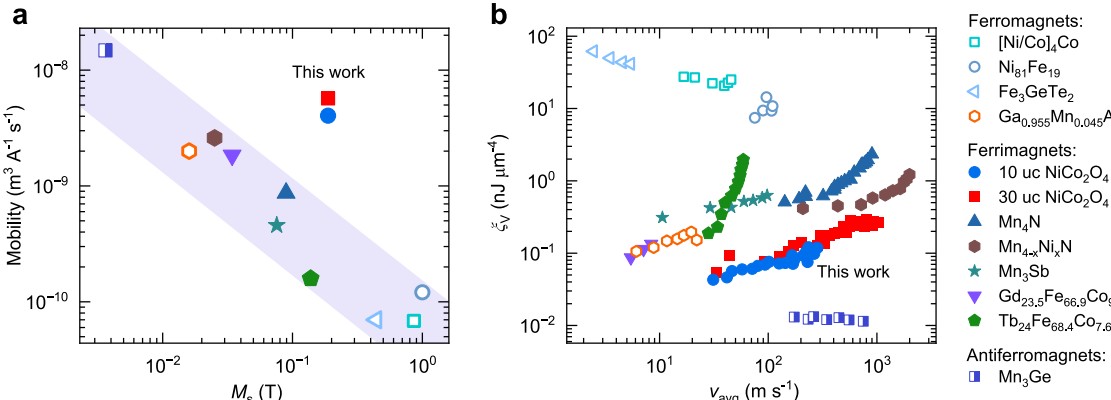

**Fig. 5 | Comparison of DW mobility and energy consumption for STT-based DW racetrack devices. a** DW mobility with respect to magnetization in ferromagnetic, ferrimagnetic, and antiferromagnetic racetracks. **b** Energy consumption per volume per DW displacement versus the gained DW velocity in these devices. Open, closed, and half-closed markers show the ferromagnets ([Co/Ni]$_4$/Co[62], Ni$_{81}$Fe$_{19}$[61], Fe$_3$GeTe$_2$[92], Ga$_{0.955}$Mn$_{0.045}$As[93]), the ferrimagnets (NiCo$_2$O$_4$ in this work, Mn$_4$N[42],

Mn$_{4-x}$Ni$_x$N[43], Mn$_3$Sb[47], Gd$_{23.5}$Fe$_{66.9}$Co$_{9.6}$[45], Tb$_{24}$Fe$_{68.4}$Co$_{7.6}$[94]), and antiferromagnets (Mn$_3$Ge[70]), respectively. All velocities were measured at room temperature, except for the Ga$_{0.955}$Mn$_{0.045}$As at 107 K, Fe$_3$GeTe$_2$ at 20 K, and Gd$_{23.5}$Fe$_{66.9}$Co$_{9.6}$ at 211 K. The results for Mn$_3$Ge, and Fe$_3$GeTe$_2$ were obtained in a single-crystal racetrack fabricated by focused ion beam, and an exfoliated nanoflake, respectively, while the others were measured in thin films.

and cross-sectional area across different material systems, regardless of device geometry. Figure 5b presents a comparative plot of $\xi_V$ as a function of $v_{avg}$ for different STT racetrack devices (see Supplementary Note 11 for alternative metrics). In general, achieving higher DW velocity comes at the expense of increased energy consumption. Overall, the performance of DW devices tends to improve from ferromagnets to antiferromagnets. Remarkably, NCO stands out as one of the highest-performing racetrack materials demonstrated in thin films, requiring ~ 0.1 nJ for a DW displacement of 1 μm in ~ 1 ns.

In conclusion, our experimental results demonstrate that the spinel oxide NCO is an excellent material platform for STT-based high-performance and high-speed DW devices. Moreover, all-optical magnetization switching using ultrafast laser pulses has recently been reported in NCO epitaxial films[89,90]. From these perspectives, ferrimagnetic spinel oxides emerge as a promising class of ferrimagnetic materials with ultrafast spin dynamics and efficient DW motion for spintronic applications.

## Methods

### Film growth and characterization

Epitaxial NCO films with thicknesses in the range of 10 uc to 30 uc (8 - 24 nm) were deposited on MAO substrates using off-axis radio frequency magnetron sputtering. The temperature was set to 320 °C and the gas (Ar:O$_2$ = 1:1) pressure was set to 100 mTorr during the film growth.

XRD spectra were measured using a SmartLab SE diffractometer (Rigaku, Japan) with out-of-plane $\theta$-$2\theta$ scanning. The general scanning range was from 35° to 110° at a scan rate of 8°/min, while a confined range from 39° to 47° was scanned at 0.5°/min for higher resolution. A cross-sectional sample was prepared using a focused ion beam for STEM images (JEOL ARM 200CF). The in-plane and out-of-plane $M$-$H$ hysteresis loops were measured using SQUID magnetometry.

### Scanning NV magnetometry

The DW structure and chirality of NCO were characterized using a scanning NV magnetometry (QZabre LLC). Magnetic stray fields near the DW region were mapped using a monolithic diamond tip containing a single NV center. The polar angle of the NV spin was 54° relative to $z$-axis. The tip stand-off distance was maintained at 227 nm, calibrated using the sample edge near the DW. A microwave antenna ( ~ 2.9 GHz) was employed for spin excitation, while spin-state detection was performed via fluorescence microscopy (excitation at 515 nm; detection around

670 nm). Line profiles were fitted to extract DW width and chirality following the method described in the Ref[54]. All the measurements were performed in ambient conditions at room temperature.

### DW racetrack preparation and MOKE microscopy imaging

The DW racetrack devices were prepared by standard photolithography using a laser writer (Heidelberg DWL66 + ), followed by reactive ion etching with Ar plasma (Plasmalab 80Plus, Oxford Instruments). The contact electrodes were connected using wire bonding. Microwave SMA cables with a bandwidth of DC – 3 GHz were used for pulse injection. The pulses were generated using a voltage pulser (UTV50P, Kentech Instruments Ltd).

The magnetic domain images were captured using a home-built wide-field MOKE microscope. A white LED (Thorlabs, MCWHLP1, wavelength: 400-750 nm) was mounted as the light source. A high-resolution digital CMOS camera (Hamamatsu, C13440-20CU) and an optical objective (× 100, Mitutoyo) were used for imaging. The pixel size was calibrated to be 0.078 μm for the captured images under this setup. The determination of the DW position was explained in Supplementary Note 4. All measurements were performed in air at room temperature.

## Data availability

The supporting data for this article is openly available from the ETH Research Collection (10.3929/ethz-b-000740603).

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

## Acknowledgements

This work was partially funded by the Swiss National Science Foundation (Grants No. 200021-236524 and 200020-212051), by the National Science Foundation of China (Grants No. 12241401), the Scientific Research Foundation of the Higher Education Institutions for Distinguished Young Scholars in Anhui Province (Grants No. 2022AH020012), and the National Key R&D Program of China (2022YFB3506000), by the facilities at Center of Free Electron Laser & High Magnetic Field (FEL&HMF) in Anhui University. A.E.K acknowledges support from the ETH Zurich Postdoctoral Fellowship Programme (24-1 FEL-038).

## Author contributions

M.W., S.D., X.C., and P.G. conceived the project. S.C., Y.Q., R.W., and X.C. grew and characterized NCO thin films. L.V.S., C.L.D., and M.W. performed NV measurements. M.W., S.D., A.D., and A.E.K. measured

domain wall dynamics. M.W., S.D., X.C., and P.G. wrote the manuscript. All authors discussed the data and commented on the manuscript.

## Competing interests

The authors declare no competing interests.
