## [Transparent Peer Review file · Nature Communications]

High-mobility inertial domain walls driven by spin-transfer torque in a ferrimagnetic spinel oxide

Corresponding Author: Dr Mingxing Wu

Version 0:

Reviewer comments:

Reviewer #1

(Remarks to the Author)

The authors studied STT driven DW motion in ferrimagnet NCO, and reported a DW velocity exceeding 1 km/s with a current density of only 2×10^{11} A/m², corresponding to a DW mobility of 5.7×10^{-9} m³/(As). They attributed it to the giant nonadiabatic STT, low magnetization, and high spin polarization in NCO. They also observed a pronounced DW inertia effect, which is used to determine material parameters. However, several critical issues have to be addressed before the work can be further considered for publication in Nat. Commun.

1. The authors use the ferromagnetic theory to interpret their key results, including the DW profile, and STT driven DW motion. They even use the same theory to extract material parameters. Given the ferrimagnetic nature and the complex crystalline structure as shown in Fig. 1(a), it is doubtful that the FM theory would give an accurate interpretation.
2. As the authors mentioned, typically the nonadiabatic STT is small in FM and can be large in FiM due to spatial variation of magnetization. This indicates the antiferromagnetic coupled sublattice plays an important role, which has been neglected in the theoretical analysis.
3. Moreover, the nonadiabatic STT coefficient β in this work is 1.4, much larger than GdFeCo (0.5). The authors should justify this and provide theoretical limit of such coefficients in different material systems.
4. In terms of the DW speed or DW mobility, 5 km/s under $J_c = 4.2 \times 10^{11}$ A/m² has been reported in FiM (GdFeCo) driven by SOT [Nat. Elec. 3, 37 (2020)]. The authors should justify why one would still choose NCO.

Other comments

1. Could the author give a theoretical limit of DW motion in NCO, which would be helpful for assessment of its potential.

Reviewer #2

(Remarks to the Author)

Key results: Race Track Memory has been studied for some time as a potential next-generation magnetic storage technology. Key factors for optimizing materials for this application include domain wall mobility and the associated energy consumption. In the submitted paper, the authors investigate domain walls in the ferrimagnetic spinel oxide NCO. They demonstrate high domain wall mobility, low energy consumption, and propose the broader class of ferrimagnetic spinel oxides as promising candidates for future applications.

Data & methodology: To my understanding, the measurements are carefully performed and the theoretical framework for data analysis is sound. The supplementary material provides ample additional information. I have no doubts about the validity of the data or the conclusions drawn.

Clarity of paper: Overall, the paper is well written and accessible, with previous literature mostly cited appropriately. There is some analogy to domain wall motion in antiferromagnets driven by SOT (Gomonay et al., Phys. Rev. Lett. 117, 017202 (2016)) and domain wall motion in ferrimagnets by magnonic currents (Donges et al., Phys. Rev. Research 2, 013293 (2020)) that the authors may wish to consider. The figures are clear and informative. However, in Fig. 4 (figure and caption), I would suggest avoiding the term "simulation" for the result shown in Fig. 4a. In my understanding, this is the outcome of an analytical calculation presented in the Supplementary Information. The term "computer simulation" by contrast, usually refers to large-scale numerical modeling. Additionally, in the supplement (line 116), "precession" should be corrected to "precession."

Originality and significance: Optimizing domain wall mobility while minimizing energy consumption is of broad relevance in spintronics. Fig. 5 convincingly demonstrates exceptionally high domain wall mobility despite the material's finite magnetization — to date, only an antiferromagnet has shown greater mobility. This is a noteworthy advance and will be of immediate interest to the spintronics community.

Conclusion: Nature Communications is dedicated to publishing high-quality research representing important advances in specific scientific disciplines. In this paper, the advance lies in the discovery of high domain wall mobility in a material with finite magnetization. I consider these findings sufficiently significant for the spintronics and materials science communities to warrant publication in Nature Communications.

Reviewer #3

(Remarks to the Author)

In this manuscript, the authors demonstrate Bloch-type domain wall (DW) velocities exceeding 1 km/s in the single-layer ferrimagnetic spinel oxide NiCo_2O_4 induced by spin-transfer torque (STT) at a current density of $2 \times 10^{11} \text{ A/m}^2$. Their exceptional DW mobility is attributed to the combination of giant nonadiabatic STT, low magnetization, and high spin polarization. Additionally, they report a pronounced DW inertia effect in this ferrimagnet due to the large nonadiabaticity of the torque. The characteristic time for DW acceleration and deceleration is $\sim 1 \text{ ns}$, shorter than that reported for typical ferromagnets. The authors' findings are sound, and the manuscript is well written. Therefore, I recommend for publication after revision of several minor points. My suggestions regarding this manuscript are as follows;

(1) The origin of large nonadiabaticity: The authors systematically elucidate the large nonadiabaticity of STT in the NiCo_2O_4 sample. However, they did not mention the physical origin of this large nonadiabaticity. Why does the NiCo_2O_4 sample exhibit such a large nonadiabaticity? In my opinion, the discussion on the origin of this large nonadiabaticity would be helpful for the reader. Please provide additional discussion on this point in revised manuscript.

(2) The relativistic effect: Recent paper [Science 370, 1438 (2020)] have demonstrated the relativistic kinematics of domain wall motion at high speed. The authors' observations also show a very high domain wall velocity up to 1 km/s. However, in Figure 3d, it appears that the domain wall velocity diverges at large applied current. What is the origin of this divergence? Is there any relativistic effect in domain wall motion at such high velocity?

(3) Minor comments: There are several minor mistakes and suggestions.

(3-1) for Figure 1: On line 97th, Figure 1c presents transmission electron microscopy (TEM) data, but the manuscript explains it as a magnetic hysteresis loop. Please provide an appropriate explanation of the TEM data.

(3-2) for Figure 3: On line 165th, Figure 4e should be corrected to Figure 3e.

(3-3) Suggestion I: In Figure 2b, please provide the scanning line of the NV center measurement results shown in Figures 2c and 2d. This will be helpful for readers.

(3-3) Suggestion II: Please include a linear fitting line in Figure 3e.

End of review report

Reviewer #4

(Remarks to the Author)

The authors of this manuscript report fast domain wall (DW) motion at velocity up to 1 km/s in NiCo_2O_4 ferrimagnetic spinel oxide under relatively low current density via spin transfer torque (STT). They attribute this rapid motion to the high spin polarization and large nonadiabaticity arising from the staggered spin configuration. In addition, they report inertial effects enabling DW motion for $\sim 1 \text{ ns}$ after turning off the pulse. While the observation of rapid domain wall motion at low current density is indeed advantageous from the viewpoint of racetrack memory, it is hard for me to think that work represent a major technological advance. Fast domain wall motion has already been reported multiple times in the literature, not only in the STT-driven cases as cited by the authors, but also in spin-orbit torque systems. It is not clear what specific advantage arises from achieving domain wall motion purely by STT, and the explanation for the fast motion does not show new physics. Moreover, the significant inertia could in fact be an obstacle for precisely controlling domain wall position using the current pulse. Overall, although the data are interesting, I am not convinced that this work meets the standards required for the publication in Nature Communications. I therefore do not recommend publication.

Below I provide several minor comments:

1. The manuscript claims that the characteristic time of $\sim 1 \text{ ns}$ is shorter than in typical ferromagnets, but a massless domain wall motion has previously been reported [J. Vogel Phys. Rev. Lett. 108, 247202 (2012)].
2. The material used here appears to have high resistivity, which is not favorable in terms of power-consumption. The authors suggested volume-normalized energy consumption, but given the relatively large film thickness used in this study, I question whether this is an appropriate parameter. A comparison in terms of energy per device length might be more relevant.
3. The domain walls appear to be quite wide (tens of nanometers), which does not seem advantageous in terms of domain density (bit-density).
4. Appropriate citations are needed for the experiments on the in-plane magnetic field dependence of domain wall velocity.
5. The fact that the domain wall moves along the current direction by STT is intriguing but is treated rather superficially. The authors should discuss whether previous studies on the directionality of STT exist, and whether alternative explanations

beyond STT are possible for the opposite STT.

6. The resistance of the sample should be specified. Given the high resistivity of the material, the resistance is likely much larger than 50Ω , raising concerns about impedance mismatch when injecting 1 ns current pulses. The inset in Fig. 4b shows a current pulse profile; it should be clarified whether this waveform was measured after transmission through the sample using an oscilloscope, or whether it simply obtained without the sample.

Version 1:

Reviewer comments:

Reviewer #1

(Remarks to the Author)

My main concern is the validity of the theoretical model. The authors justified that they use the 1D model with antiferromagnetically coupled sublattices, which is an unsatisfied answer. The same model has been extensively used in the previous studies of ferrimagnets such as CoTb, GdFeCo, or antiferromagnets. Therefore, nothing is new if the authors rely on such simplified model to explain the experimental results, and there is no reason to believe NCO has new physics compared to other ferrimagnets or antiferromagnets. If the authors believe there is new physics in NCO, I suggest the author to seriously consider more dedicated theories that can capture the unique characteristics of NCO, such as the complicated lattice structure and exchange interactions beyond the nearest neighbor coupling. At this stage, I do not recommend its publication in nature communications.

Reviewer #2

(Remarks to the Author)

Key point of the submitted manuscript is the exceptionally high domain wall mobility driven by spin-transfer torque (STT). My previous report was overall positive, and the authors' replies to my remarks as well as the corresponding changes in the revised manuscript are satisfactory. After reviewing the other referees' reports, I noted that Referee 4 raised serious concerns regarding the novelty of the work, claiming that high domain wall mobilities have been reported before — however, without providing any specific evidence. Referee 1 (in remark #4) compares the present results seriously to a previous study that demonstrated similarly high domain wall mobilities, but in a very different system and driven by spin-orbit torque (SOT), a distinct mechanism. I find the authors' response to this remark convincing. In conclusion, I recommend publication of the manuscript in Nature Communications.

Reviewer #3

(Remarks to the Author)

I would like to express my gratitude to the authors for their tremendous efforts in alleviating my concerns. Therefore, I am pleased to recommend the publication of this manuscript.

Reviewer #4

(Remarks to the Author)

The authors have responded well to the reviewers' comments and have provided persuadable answers in several aspects. While I believe that the NCO material presents meaningful results in terms of the development of domain-wall-motion-based devices, I still question whether this is a significant advance. Although spin-transfer torque (STT) in ferrimagnets has not been extensively studied, I do not think the results present a significant advance compared to previous ferrimagnet studies, which required an external magnetic field to achieve fast domain-wall motion. The need for an external field is not a major issue, as it can be easily expected if the Dzyaloshinskii–Moriya interaction (DMI) is tuned to stabilize a Néel wall. While the introduction of a new material platform is certainly of interest and could stimulate further studies, however, because the current findings can still be explained within the scope of existing models, it is hard for me to agree that this work 'improve the fundamental understanding of phenomena leading to high DW mobilities.' By the way, I think presenting the longitudinal field H_x dependence would constitute a genuine new finding. Since non-adiabaticity is inversely proportional to the domain-wall width (in terms of mistracking), verifying whether the domain-wall mobility decreases as H_x modifies the wall width would significantly enhance the novelty of the work.

Version 2:

Reviewer comments:

Reviewer #2

(Remarks to the Author)

In my last report, I had no further remarks or questions and I recommended publication of the manuscript in Nature Communications. Having read the other reports, I note a disagreement between Referee 1 and the authors concerning the

modeling. Referee 1 concludes that "... nothing is new if the authors rely on such a simplified model to explain the experimental results ...". On this point, I disagree and find the authors' response convincing. In particular, a new experimental finding does not necessarily require a new type of theory. Identifying a material with exceptional parameters within an established model can still represent a high degree of novelty. Overall, I continue to recommend publication.

Reviewer #3

(Remarks to the Author)

Dear authors,

I am already convinced by the authors' responses in the previous round, and I recommended that the results be published in Nature Communications. In this round as well, other reviewers posed sharp questions, and I believe the authors provided convincing answers. Once again, I recommend publication.

Reviewer #4

(Remarks to the Author)

First, the referee thanks the authors for their response regarding the additional experiments. The authors have experimentally verified the dependence of the domain wall (DW) velocity v on H_x and H_y as predicted by the 1D STT model. Nevertheless, even with this confirmation, and even if the authors argue that the application of the 1D model still allows them to extract important material parameters of NCO, I believe that significant limitations remain.

The authors emphasize that, based on the application of the 1D model, NCO exhibits a very large ratio $\beta/\alpha \sim 20$ and an exceptionally large nonadiabaticity $\beta \sim 1.4$. They claim that the observation of such a large β in NCO, compared to the previously studied GdFeCo ferrimagnet with $\beta \sim 0.5$, constitutes novelty. However, I believe that this interpretation potentially holds a fundamental issue arising from treating a two-sublattice ferrimagnetic system within a ferromagnetic 1D model.

According to a study on the damping parameter α in GdFeCo ferrimagnets (PRL 122, 127203 (2019)), the effective α in ferrimagnets can be significantly smaller than the value inferred from ferromagnetic FMR analysis. In the present work, the authors use $\alpha = 0.08$ for NCO. If this value is also subjected the same limitation—namely, interpreting a ferrimagnet as a ferromagnet, as implicitly done in the magnetization treatment within the 1D model—then the resulting analysis could change substantially.

Since the domain wall velocity scales with β/α , if the actual α is smaller than the assumed value, the resultant nonadiabaticity $\beta \sim 1.4$ may no longer be exceptionally large. Furthermore, while the authors claim $\beta/\alpha \sim 17-20$ for NCO, one the value is nearly 100 for GdFeCo from Ref. [Nat. Electron. 2, 389–393 (2019)].

Of course, domain wall motion at high velocity under low current density is an important result from an applications perspective. However, I find it difficult to agree with the authors' arguments that the application of the 1D model alone is sufficient to firmly establish the superior intrinsic properties of NCO—such as its claimed large nonadiabaticity and high spin polarization. For these reasons, I remain hesitant to recommend publication.

made.

Response Letter

We sincerely appreciate the four reviewers' thoughtful comments. The valuable suggestions helped us to improve the manuscript and stimulated further discussion of our results. We have carefully addressed all the reviewers' comments with a point-by-point response below, and revised our manuscript (highlighted in blue) accordingly. A summary of changes in the revised manuscript are provided at the end of this letter.

Reviewer #1

(Remarks to the Author):

The authors studied STT driven DW motion in ferrimagnet NCO, and reported a DW velocity exceeding 1 km/s with a current density of only 2×10^{11} A/m², corresponding to a DW mobility of 5.7×10^{-9} m³/(As). They attributed it to the giant nonadiabatic STT, low magnetization, and high spin polarization in NCO. They also observed a pronounced DW inertia effect, which is used to determine material parameters. However, several critical issues have to be addressed before the work can be further considered for publication in Nat. Commun.

Response: We thank the reviewer for providing critical comments on our work and pointing out issues that need further consideration. We address the comments and give point-by-point responses as follows.

1. The authors use the ferromagnetic theory to interpret their key results, including the DW profile, and STT driven DW motion. They even use the same theory to extract material parameters. Given the ferrimagnetic nature and the complex crystalline structure as shown in Fig. 1(a), it is doubtful that the FM theory would give an accurate interpretation.

Response: In this manuscript, we employed the one-dimensional (1D) model to describe STT-driven DW dynamics in ferrimagnetic NCO. The 1D model is derived from the Landau–Lifshitz–Gilbert (LLG) equation. The validity of this model for ferrimagnets has been theoretically established^{1,2} and experimentally verified^{3,4,5,6} in multiple studies.

As the reviewer noted, ferrimagnets possess antiferromagnetically coupled sublattices, which must be explicitly considered when deriving DW dynamics. Accordingly, the 1D model has been extended to ferrimagnets by incorporating the sublattice magnetic moments (M_1 and M_2). The generic expression of the DW velocity below the Walker breakdown gives: $v = \frac{\beta}{L_\alpha} L_S u$, where $L_S = \frac{M_1}{\gamma_1} - \frac{M_2}{\gamma_2}$ is the angular momentum density ($\gamma = \frac{g\mu_B}{\hbar}$), and $L_\alpha = \alpha L_S$, $L_S u = \frac{\hbar P}{2e} j$ [Table 1 of Ref. ¹]. In this formulation, α , β , P represent the effective damping, nonadiabatic torque parameter, and spin polarization, defined to consistently capture the sublattice contributions.

In general, ferrimagnets can be classified into two categories depending on whether a compensation temperature exists, and their analytical descriptions of DW dynamics can differ accordingly.

¹ Haltz, E. *et al.* Domain wall dynamics in antiferromagnetically coupled double-lattice systems, *Phys. Rev. B* **103**, 014444 (2021). DOI: <https://doi.org/10.1103/PhysRevB.103.014444>.

² Jing, K. Y. *et al.* Current-driven domain wall motion in ferrimagnetic nanowires. *Phys. Rev. B*, **110**, 054414 (2024). DOI: <https://doi.org/10.1103/PhysRevB.110.054414>.

³ Okuno, T. *et al.* Spin-transfer torques for domain wall motion in antiferromagnetically coupled ferrimagnets. *Nat Electron* **2**, 389–393 (2019). DOI: <https://doi.org/10.1038/s41928-019-0303-5>.

⁴ Thomas, L. *et al.*, Dynamics of magnetic domain walls under their own inertia. *Science* **330**, 1810–1813 (2010). DOI: [10.1126/science.1197468](https://doi.org/10.1126/science.1197468).

⁵ Siddiqui, S. *et al.*, Current-induced domain wall motion in a compensated ferrimagnet. *Phys. Rev. Lett.* **121**, 057701 (2018). DOI: <https://doi.org/10.1103/PhysRevLett.121.057701>.

⁶ Gushi, T. *et al.*, Large current driven domain wall mobility and gate tuning of coercivity in ferrimagnetic Mn₄N thin films. *Nano Letters* **19**, 8716–8723 (2019). DOI: [10.1021/acs.nanolett.9b03416](https://doi.org/10.1021/acs.nanolett.9b03416).

In the case of rare-earth transition-metal ferrimagnets such as GdFeCo ^{7,8}, $\text{Gd}_x\text{Co}_{1-x}$ ^{9,10} and $\text{Co}_{1-x}\text{Tb}_x$ ¹¹, the magnetization compensation temperature (T_M) occurs below room temperature due to the distinct temperature dependencies of the sublattice magnetizations (see left panel of Fig. R1, data from Ref.¹⁰). In addition, the Landé g -factors differ between rare-earth and transition-metal atoms, e.g., $g_{\text{Gd}} = 2$ and $g_{\text{Co}} = 2.2$. This leads to distinct gyromagnetic ratios and, consequently, an angular momentum compensation temperature T_A ($L_S = 0$) away from T_M . In this case, the DW velocity $v = \frac{\beta}{\alpha L_S} \frac{\hbar P}{2e} j$ diverges as the temperature approaches T_A . Only when the temperature is far away from T_A , $L_S \approx \frac{M_1 - M_2}{\gamma} = \frac{M_S}{\gamma}$ with $\gamma = \gamma_1 \approx \gamma_2$ and $M_S = M_1 - M_2$. The DW velocity becomes $v = \frac{\beta}{\alpha} \frac{g \mu_B P}{2e M_S} j$, similar to ferromagnets.

In NCO, the magnetic moments are carried by eight Ni ions ($1.5 \mu_B/\text{Ni}$) on octahedral sites and eight Co ions ($3.5 \mu_B/\text{Co}$) on tetrahedral sites, which are antiferromagnetically aligned. The sublattice magnetic moment of Co remains larger than that of Ni at all temperatures below the Curie temperature, and thus no magnetization compensation point is reached. This can be verified from the M - T and H_c - T curves, where no dip or peak emerges, as shown in the right panel of Fig. R1. Moreover, Co and Ni possess nearly identical Landé g -factors (≈ 2.2), meaning that the angular momentum does not compensate either. As a result, L_S can be directly simplified as $L_S = \frac{M_1 - M_2}{\gamma} = \frac{M_S}{\gamma}$, which does not diverge in NCO ($L_S \neq 0$). DW velocity is thus expressed as: $v = \frac{\beta}{\alpha} \frac{g \mu_B P}{2e M_S} j$. This formulation is equivalent to that of ferromagnets in the whole temperature range. Therefore, for

⁷ Kim, K.J. *et al.* Fast domain wall motion in the vicinity of the angular momentum compensation temperature of ferrimagnets. *Nat Mater* **16**, 1187–1192 (2017). DOI: <https://doi.org/10.1038/nmat4990>.

⁸ Okuno, T. *et al.* Spin-transfer torques for domain wall motion in antiferromagnetically coupled ferrimagnets. *Nat Electron* **2**, 389–393 (2019). DOI: <https://doi.org/10.1038/s41928-019-0303-5>.

⁹ Caretta, L. *et al.* Fast current-driven domain walls and small skyrmions in a compensated ferrimagnet. *Nature Nanotech* **13**, 1154–1160 (2018). DOI: <https://doi.org/10.1038/s41565-018-0255-3>.

¹⁰ Cai, K. *et al.* Ultrafast and energy-efficient spin-orbit torque switching in compensated ferrimagnets. *Nat Electron* **3**, 37–42 (2020). DOI: <https://doi.org/10.1038/s41928-019-0345-8>.

¹¹ Siddiqui, S. *et al.*, Current-induced domain wall motion in a compensated ferrimagnet. *Phys. Rev. Lett.* **121**, 057701 (2018). DOI: <https://doi.org/10.1103/PhysRevLett.121.057701>.

ferrimagnets without compensation point—such as NCO, Bi-YIG¹², Mn₄N¹³— the 1D model can be directly applied to describe their DW dynamics.

Figure R1. left: M - T and H_c - T curves for GdCo with compensation points (data from the Ref. ⁹). Right: M - T and H_c - T curves for NCO showing the absence of compensation.

Regarding the DW profile, the NV center is sensitive only to the stray field near the surface generated by the magnetization within DW. This technique is independent of the type of sublattice coupling and is applicable to DWs in ferromagnetic¹⁴, ferrimagnetic¹⁵, and antiferromagnetic materials^{16,17}. In these studies, the same approach has been employed to extract DW width and chirality, consistent with Eq. (1) of our manuscript.

In summary, for the materials that exhibit a compensation point, the ferrimagnetic DW dynamics become more complex only when the temperature approaches T_A , where the DW velocity tends to diverge. Away from the compensation point, the expression for DW dynamics becomes equivalent to that of ferromagnets. For the materials that do not show compensation, the description

¹² Caretta, L. *et al.*, Relativistic kinematics of a magnetic soliton. *Science* **370**, 1438-1442 (2020). DOI: [10.1126/science.aba5555](https://doi.org/10.1126/science.aba5555).

¹³ Gushi, T. *et al.*, Large current driven domain wall mobility and gate tuning of coercivity in ferrimagnetic Mn₄N thin films. *Nano Letters* **19**, 8716-8723 (2019). DOI: [10.1021/acs.nanolett.9b03416](https://doi.org/10.1021/acs.nanolett.9b03416).

¹⁴ Tetienne, JP. *et al.* The nature of domain walls in ultrathin ferromagnets revealed by scanning nanomagnetometry. *Nat Commun* **6**, 6733 (2015). DOI: <https://doi.org/10.1038/ncomms7733>.

¹⁵ Vélez, S. *et al.* High-speed domain wall racetracks in a magnetic insulator. *Nat Commun* **10**, 4750 (2019). DOI: <https://doi.org/10.1038/s41467-019-12676-7>.

¹⁶ Hedrich, N. *et al.* Nanoscale mechanics of antiferromagnetic domain walls. *Nat. Phys.* **17**, 574–577 (2021). DOI: <https://doi.org/10.1038/s41567-020-01157-0>.

¹⁷ Wörnle, M. S. *et al.* Coexistence of Bloch and Néel walls in a collinear antiferromagnet. *Phys. Rev. B* **103**, 094426 (2021). DOI: <https://doi.org/10.1103/PhysRevB.103.094426>.

of ferrimagnetic DW motion remains effectively identical to that of ferromagnetic DWs over the entire temperature range. Therefore, we believe that our theoretical framework is appropriate for describing DW dynamics for NCO and is consistent with previous treatments reported for materials such as in Bi-YIG¹⁸, Mn₄N¹⁹, and GdFeCo²⁰.

In response to this comment, we added a sentence to the main text in line #172, and a new section in the Supplementary Material (**Supplementary Note 7: DW dynamics formulation for ferrimagnetic NCO**) including a discussion of the DW velocity in ferrimagnets close and away from compensation and the data supporting our conclusions for NCO in Fig. R1.

¹⁸ Caretta, L. *et al.* Relativistic kinematics of a magnetic soliton. *Science* **370**, 1438-1442 (2020). DOI: [10.1126/science.aba5555](https://doi.org/10.1126/science.aba5555).

¹⁹ Gushi, T. *et al.* Large current driven domain wall mobility and gate tuning of coercivity in ferrimagnetic Mn₄N thin films. *Nano Letters* **19**, 8716-8723 (2019). DOI: [10.1021/acs.nanolett.9b03416](https://doi.org/10.1021/acs.nanolett.9b03416).

²⁰ Okuno, T. *et al.* Spin-transfer torques for domain wall motion in antiferromagnetically coupled ferrimagnets. *Nat Electron* **2**, 389–393 (2019). DOI: <https://doi.org/10.1038/s41928-019-0303-5>.

2. As the authors mentioned, typically the nonadiabatic STT is small in FM and can be large in FiM due to spatial variation of magnetization. This indicates the antiferromagnetic coupled sublattice plays an important role, which has been neglected in the theoretical analysis.

Response: We thank the reviewer for this interesting comment. Indeed, the antiferromagnetic spin alignment in the sublattice of a FiM is expected to play an important role in generating a strong nonadiabatic torque. However, the presence of antiferromagnetic coupling does not invalidate the DW description used for a single-lattice material, in terms of effective parameters. In fact, away from momentum compensation, the stronger the antiferromagnetic coupling the more the description of DW dynamics in terms of a single effective lattice is justified, as the antiferromagnetic spins are rigidly coupled together.

Nonadiabatic STT arises when conduction electron spins cannot perfectly follow the local magnetization as they travel through a magnetic texture (e.g., a domain wall) [Fig. 6 of the Ref.²¹]. This misalignment creates an additional torque term $\boldsymbol{\tau}_{\text{NA}} = \beta \mathbf{m} \times [(\mathbf{u} \cdot \nabla) \mathbf{m}]$, which is proportional to the spatial variation of magnetization [Eq. 3 of Ref.²²]. To this end, the role of antiferromagnetic sublattice moments is to cause strong mistracking of electron spins, thereby inducing a large β .

As elaborated in comment 1, by considering two individual LLG equations for the two sublattices, in the absence of a compensation point, the DW velocity is given by $= \frac{\beta}{\alpha} \frac{g u_B P}{2 e M_s} j$, as in a ferromagnet, but expressed in terms of the effective material parameters. In typical ferromagnets, $\beta \ll \alpha$ because of the slow spatial variation of the magnetization. In contrast, in ferrimagnets and antiferromagnets the effective β can be much larger than α .

²¹ Tatara, G. *et al.* Microscopic approach to current-driven domain wall dynamics. *Phys. Rep.* **468**, 213–301 (2008). DOI: <https://doi.org/10.1016/j.physrep.2008.07.003>.

²² Thiaville, A. *et al.* Micromagnetic understanding of current-driven domain wall motion in patterned nanowires. *Europhys. Lett.* **69**, 990–996 (2005). DOI: [10.1209/epl/i2004-10452-6](https://doi.org/10.1209/epl/i2004-10452-6).

3. Moreover, the nonadiabatic STT coefficient β in this work is 1.4, much larger than GdFeCo (0.5). The authors should justify this and provide theoretical limit of such coefficients in different material systems.

Response: We thank the reviewer for the suggestion. The β coefficient can be derived from the 1D model as: $\beta = \left(\frac{\lambda_J}{\lambda_{sf}}\right)^2$ [Eq. 7 of Ref. ²³]. Here $\lambda_J = \sqrt{\frac{2\hbar D_0}{J}}$ and $\lambda_{sf} = \sqrt{2D_0\tau_{sf}}$ are the exchange length and spin-flip length, where D_0 , J , and τ_{sf} are the spin diffusion constant, s - d exchange interaction energy, and spin-flip relaxation time. These parameters are material dependent which may make different β values in NCO and GdFeCo. In addition, the value of β for NCO is extracted from velocity – current density curve as well as DW inertia effect, whereas for GdFeCo it is obtained by fitting the DW mobility across the compensation temperature. The use of different approaches may introduce uncertainties, which could account for the relatively large β for NCO.

The key to theoretically determining β coefficient lies in the estimation of material parameters. Precisely quantifying them is challenging, while we phenomenologically compare their order of magnitude for the different material systems. We first consider the λ_J , which is governed by J . The s - d exchange originates from the interaction between itinerant s electrons and localized d electrons. It is generally independent of the type of sublattices coupling. Hence, we adopt a representative value of $\lambda_J = 1$ nm from the literature [Ref. ²³]. In contrast, λ_{sf} depends strongly on the sublattice coupling, as spin-flip process is highly sensitive to the magnetic structure. The staggered antiferromagnetic sublattice induces the faster spin dephasing than ferromagnetic sublattice, resulting in a shorter λ_{sf} . Recent theoretical calculation has revealed a rapidly reduced spin-flip time when transitioning from a ferromagnetic to an antiferromagnetic sublattice [Ref.²⁴]. For a ferromagnetic sublattice, λ_{sf} is generally on the order of several nanometers²⁵, whereas in antiferromagnet it can be as short as 0.5 nm²⁶. Accordingly, we adopt $\lambda_{sf} = 5$ nm for the

²³ Thiaville, A. *et al.* Micromagnetic understanding of current-driven domain wall motion in patterned nanowires. *Europhys. Lett.* **69**, 990–996 (2005). DOI: [10.1209/epl/i2004-10452-6](https://doi.org/10.1209/epl/i2004-10452-6).

²⁴ Lu, H. *et al.* Magnetic structure-dependent ultrafast spin relaxation in magnet CrI₃: a time-domain ab initio study. *Nano Letters* **24**, 8940-8947 (2024). DOI: [10.1021/acs.nanolett.4c01809](https://doi.org/10.1021/acs.nanolett.4c01809).

²⁵ Bass, J. *et al.* Spin-diffusion lengths in metals and alloys, and spin-flipping at metal/metal interfaces: an experimentalist's critical review. *J. Phys.: Condens. Matter* **19**, 183201 (2007). DOI: [10.1088/0953-8984/19/18/183201](https://doi.org/10.1088/0953-8984/19/18/183201).

²⁶ Zhang, W. *et al.* Spin Hall effects in metallic antiferromagnets. *Phys. Rev. Lett.* **113**, 196602 (2014). DOI: <https://doi.org/10.1103/PhysRevLett.113.196602>.

ferromagnetic lattice²³, and $\lambda_{sf} = 0.5$ nm for the antiferromagnetic lattice²⁶. Using these values, we obtain $\beta = 0.04$ for the ferromagnetic sublattice, and $\beta = 4$ for the antiferromagnetic sublattice. The estimated β is in 10^{-2} order for a ferromagnet, consistent with general expectations, whereas in an antiferromagnet it can be up to two orders of magnitude larger, owing to the strong mistracking of electron spins when traveling through antiferromagnetic sublattice.

In response to the reviewer's comments 2 and 3, we modified the **Discussion Section** and added the above discussion on the β coefficient in Line # 222–242 in the revised manuscript (text highlighted in blue).

4. In terms of the DW speed or DW mobility, 5 km/s under $J_c=4.2e11$ A/m² has been reported in FiM (GdFeCo) driven by SOT [Nat. Elec. 3, 37 (2020)]. The authors should justify why one would still choose NCO.

Response: We thank the reviewer for this valuable comment. We agree that fast DW motion with low current density has been demonstrated in GdCo²⁷. Nevertheless, we emphasize that the search for materials with high DW velocity should not be restricted to SOT systems or a single material class. From a fundamental point of view, the observation of large nonadiabatic STT in a spinel oxide shows that nonadiabatic spin transport is a general feature of ordered materials with antiferromagnetic spin coupling. From a practical point of view, DW motion in NCO has comparative advantages relative to other systems.

First, NCO enables fast DW motion in the absence of any external magnetic field, whereas GdCo requires the application of an in-plane magnetic field of 1.4 kOe to achieve fast DW motion.

Second, the driving mechanism in NCO is STT, whereas the cited work investigated SOT-driven DW motion. As we explained in the first paragraph of the manuscript, STT offers several complementary features relative to SOT: (i) it enables DW motion in a single-layer magnetic structure, eliminating the need for a heavy-metal spin source; (ii) it is effective for both Néel- and Bloch-type DWs without requiring a bias magnetic field; and (iii) the direction of DW motion can be easily controlled by the polarity of the applied current irrespective of the DW chirality.

The main obstacle of STT is its lower efficiency compared to SOT^{28,29}, as it typically requires relatively high current densities, especially for pulse durations on the order of 1 ns. This limitation motivated our investigation into strategies for enhancing STT efficiency, as reported in this work.

According to the 1D model, the STT-driven DW velocity $v = \frac{\beta}{\alpha} \frac{g\mu_B P}{2eM_s} j$. This expression indicates that DW mobility can be enhanced by selecting materials with large spin polarization (P), small saturation magnetization (M_s), and large nonadiabaticity (β). In conventional ferromagnets,

²⁷ Cai, K. *et al.* Ultrafast and energy-efficient spin-orbit torque switching in compensated ferrimagnets. *Nat Electron* **3**, 37–42 (2020). DOI: <https://doi.org/10.1038/s41928-019-0345-8>.

²⁸ Churemart, J. *et al.* Current-induced domain wall motion: Comparison of STT and SHE. *J. Magn. Magn. Mater.* **529**, 167838 (2021). DOI: <https://doi.org/10.1016/j.jmmm.2021.167838>.

²⁹ Caretta, L. *et al.* Domain walls speed up in insulating ferrimagnetic garnets. *APL Mater.* **12**, 011106 (2024). DOI: <https://doi.org/10.1063/5.0159669>.

spin polarization and magnetization are typically correlated, making it challenging to simultaneously achieve large P and small M_s . However, certain ferrimagnets, such as the Heusler alloy Mn_3Ge and the spinel oxide NiCo_2O_4 , possess unique band structures near the Fermi level that allow high spin polarization despite a relatively small magnetization. Moreover, ferrimagnets with antiferromagnetically coupled sublattices are expected to exhibit enhanced nonadiabaticity. Motivated by these considerations, we selected NCO as a promising platform to investigate STT-driven DW dynamics and found that STT can achieve an efficiency comparable to that of SOT.

In response to the reviewer's comment, we have revised **Introduction Section** to improve the story flow. The new introduction now follows a clear structure: first, outlining the advantages of STT; second, presenting strategies for enhancing STT efficiency; and finally, explaining the rationale for choosing NCO as the material system (text highlighted in blue in the Introduction Section).

Other comments

1. Could the author give a theoretical limit of DW motion in NCO, which would be helpful for assessment of its potential.

Response: We thank the reviewer for this interesting comment. In the case of antiferromagnetically coupled sublattices, owing to the strong exchange field, the upper velocity limit of DW is governed by the magnon group velocity. This can be derived from the spin-wave dispersion relation, and is given by $v_g^{\max} = \frac{2A}{dS}$, where A is the exchange stiffness, d the lattice constant and $S = |S_1| + |S_2|$ the total angular momentum^{30,31,32}.

To estimate the velocity limit in NCO, we take $d = 0.8$ nm. A can be estimated from the DW width: $\delta = \pi \sqrt{\frac{A}{K_u}}$. Taking $\delta = 39 \pm 13$ nm from our NV measurements, $K_u = 0.2$ MJ/m³ [from Ref. ³³], we obtain $A = 3.2 \pm 2.2 \times 10^{-11}$ J/m. The magnetic moments in NCO are carried by antiferromagnetically coupled Ni ($1.5 \mu_B/\text{Ni}$) and Co ($3.5 \mu_B/\text{Co}$) sublattice [Fig. 2 of Ref. ³⁴]. Using $g_{\text{Co}} = g_{\text{Ni}} = 2.2$, we obtain $S = \frac{5\mu_B}{\gamma}/\text{f. u.} = 3.62 \times 10^{-6}$ kg/(m · s). Based on these parameters, we estimate the theoretical limit of DW velocity as $v_g^{\max} = 22 \pm 15$ km/s. The large uncertainty originates from the multiple measurements of DW width using NV magnetometry.

However, achieving such a velocity limit requires a current density as high as 4×10^{12} A/m² according to the linear expansion of Fig. 3e. Although it remains challenging at present, future strategies, such as reducing magnetization (e.g., through Ni doping) and increasing film conductivity, could further enhance DW mobility, potentially enabling the observation of relativistic DW dynamics in NCO.

³⁰ Shiino, T. *et al.* Antiferromagnetic domain wall motion driven by spin-orbit torques. *Phys. Rev. Lett.* **117**, 087203 (2016). DOI: <https://doi.org/10.1103/PhysRevLett.117.087203>.

³¹ Gomonay, O *et al.* High antiferromagnetic domain wall velocity induced by Néel spin-orbit torques. *Phys. Rev. Lett.* **117**, 017202 (2016). DOI: <https://doi.org/10.1103/PhysRevLett.117.017202>.

³² Caretta, L. *et al.*, Relativistic kinematics of a magnetic soliton. *Science* **370**, 1438-1442 (2020). DOI: [10.1126/science.aba5555](https://doi.org/10.1126/science.aba5555).

³³ Shen, Y. *et al.* Tuning of ferrimagnetism and perpendicular magnetic anisotropy in NiCo₂O₄ epitaxial films by the cation distribution. *Phys. Rev. B* **101**, 094412 (2020). DOI: <https://doi.org/10.1103/PhysRevB.101.094412>.

³⁴ Xu, X *et al.* Epitaxial NiCo₂O₄ film as an emergent spintronic material: Magnetism and transport properties. *J. Appl. Phys.* **132**, 020901 (2022). DOI: <https://doi.org/10.1063/5.0095326>.

In response to this comment, we added a new section in the Supplementary Material (**Supplementary Note 6: Theoretical limit of DW velocity in NCO**) including the above discussion of the theoretical velocity limit of DW motion in NCO.

Reviewer #2

(Remarks to the Author):

1. Key results: Race Track Memory has been studied for some time as a potential next-generation magnetic storage technology. Key factors for optimizing materials for this application include domain wall mobility and the associated energy consumption. In the submitted paper, the authors investigate domain walls in the ferrimagnetic spinel oxide NCO. They demonstrate high domain wall mobility, low energy consumption, and propose the broader class of ferrimagnetic spinel oxides as promising candidates for future applications.

Data & methodology: To my understanding, the measurements are carefully performed and the theoretical framework for data analysis is sound. The supplementary material provides ample additional information. I have no doubts about the validity of the data or the conclusions drawn.

Response: We thank the reviewer for the thoughtful and positive evaluation of our work.

2. Clarity of paper: Overall, the paper is well written and accessible, with previous literature mostly cited appropriately. There is some analogy to domain wall motion in antiferromagnets driven by SOT (Gomonay et al., *Phys. Rev. Lett.* **117**, 017202 (2016)) and domain wall motion in ferrimagnets by magnonic currents (Donges et al., *Phys. Rev. Research* **2**, 013293 (2020)) that the authors may wish to consider. The figures are clear and informative. However, in Fig. 4 (figure and caption), I would suggest avoiding the term “simulation” for the result shown in Fig. 4a. In my understanding, this is the outcome of an analytical calculation presented in the Supplementary Information. The term “computer simulation” by contrast, usually refers to large-scale numerical modeling. Additionally, in the supplement (line 116), “procession” should be corrected to “precession.”

Response: In response to these suggestions, we have made the following modifications in the revised manuscript:

1. We have included the pioneer theoretical works when introducing the SOT for DW motion in the revised manuscript in Ref. [31] and [32]:

[31] Gomonay, O., Jungwirth, T. & Sinova, J. High antiferromagnetic domain wall velocity induced by Néel spin-orbit torques. *Phys. Rev. Lett.* **117**, 017202 (2016).

[32] Shiino, T., Oh, S., Haney, P., Lee, S., Go, G., Park, B. & Lee, K. Antiferromagnetic domain wall motion driven by spin-orbit torques. *Phys. Rev. Lett.* **117**, 087203 (2016).

2. We have introduced magnonic currents as an additional mechanism for DW motion, alongside STT and SOT, with properly cited Refs. [33–37]:

[33] Fan, Y., Gross, M. J., Fakhrul, T., Finley, J., Hou, J. T., Ngo, S., Liu, L. & Ross, C. A. Coherent magnon-induced domain-wall motion in a magnetic insulator channel. *Nat Nanotechnol* **18**, 1000–1004 (2023).

[34] Wang, W., Albert, M., Beg, M., Bisotti, M. A., Chernyshenko, D., Cortés-Ortuño, D., Hawke, I. & Fangohr, H. Magnon-driven domain-wall motion with the Dzyaloshinskii-Moriya interaction. *Phys. Rev. Lett.* **114**, 087203 (2015).

[35] Wang, X. G., Guo, G. H., Nie, Y. Z., Zhang, G. F. & Li, Z. X. Domain wall motion induced by the magnonic spin current. *Phys. Rev. B* **86**, 054445 (2012).

[36] Donges, A., Grimm, N., Jakobs, F., Selzer, S., Ritzmann, U., Atxitia, U., & Nowak, U. Unveiling domain wall dynamics of ferrimagnets in thermal magnon currents: Competition of angular momentum transfer and entropic torque. *Phys. Rev. Res* **2**, 013293 (2020).

[37] Kim, K. W., Lee, S. W., Moon, J. H., Go, G., Manchon, A., Lee, H. W., Everschor-Sitte, K. & Lee, K. J. *Phys Rev Lett* **122**, 147202 (2019).

3. We have removed the improper description of “simulation”, “experiment” in Fig. 4 and captions in the revised manuscript.
4. We have corrected the typo of “procession” in the Supplementary Material in the revised manuscript.

3. Originality and significance: Optimizing domain wall mobility while minimizing energy consumption is of broad relevance in spintronics. Fig. 5 convincingly demonstrates exceptionally high domain wall mobility despite the material's finite magnetization — to date, only an antiferromagnet has shown greater mobility. This is a noteworthy advance and will be of immediate interest to the spintronics community.

Conclusion: Nature Communications is dedicated to publishing high-quality research representing important advances in specific scientific disciplines. In this paper, the advance lies in the discovery of high domain wall mobility in a material with finite magnetization. I consider these findings sufficiently significant for the spintronics and materials science communities to warrant publication in Nature Communications.

Response: We appreciate the reviewer's remarks on the significance of our work. This encouragement motivates us to continue exploring novel material systems and the nonadiabatic STT mechanism, which we believe will contribute to the advancement of STT-based DW racetrack memory devices.

Reviewer #3

(Remarks to the Author):

In this manuscript, the authors demonstrate Bloch-type domain wall (DW) velocities exceeding 1 km/s in the single-layer ferrimagnetic spinel oxide NiCo_2O_4 induced by spin-transfer torque (STT) at a current density of $2 \times 10^{11} \text{ A/m}^2$. Their exceptional DW mobility is attributed to the combination of giant nonadiabatic STT, low magnetization, and high spin polarization. Additionally, they report a pronounced DW inertia effect in this ferrimagnet due to the large nonadiabaticity of the torque. The characteristic time for DW acceleration and deceleration is ~ 1 ns, shorter than that reported for typical ferromagnets. The authors' findings are sound, and the manuscript is well written. Therefore, I recommend for publication after revision of several minor points. My suggestions regarding this manuscript are as follows;

Response: We appreciate the reviewer's comments on the soundness of our results and the overall positive evaluation of our work. We have carefully considered the reviewer's comments and revised the manuscript accordingly.

1. The origin of large nonadiabaticity: The authors systematically elucidate the large nonadiabaticity of STT in the NiCo₂O₄ sample. However, they did not mention the physical origin of this large nonadiabaticity. Why does the NiCo₂O₄ sample exhibit such a large nonadiabaticity? In my opinion, the discussion on the origin of this large nonadiabaticity would be helpful for the reader. Please provide additional discussion on this point in revised manuscript.

Response: We thank the reviewer for this thoughtful comment, which was also raised by Reviewer 1. Nonadiabatic STT arises when the conduction electron spins cannot perfectly follow the local magnetization as they travel through a magnetic texture (e.g., a domain wall) [Fig. 6 of the Ref.³⁵]. This misalignment creates an additional torque term $\boldsymbol{\tau}_{\text{NA}} = \beta \mathbf{m} \times [(\mathbf{u} \cdot \nabla) \mathbf{m}]$, which is proportional to the spatial variation of magnetization [Eq. 3 of Ref. ³⁶]. Therefore, in typical ferromagnets, $\beta \ll \alpha$ because of the slow spatial variation of magnetization in a DW. In contrast, the effective β can be much larger than α in ferrimagnets and antiferromagnets thanks to the antiferromagnetic alignment of neighboring spin moments.

The quantity of β coefficient can be derived from the one-dimensional model as: $\beta = \left(\frac{\lambda_J}{\lambda_{sf}}\right)^2$. Here $\lambda_J = \sqrt{\frac{2\hbar D_0}{J}}$ and $\lambda_{sf} = \sqrt{2D_0\tau_{sf}}$ are the exchange length and spin-flip length, where D_0 , J , and τ_{sf} are the spin diffusion constant, s - d exchange interaction energy, and spin-flip relaxation time [Eq. 7 of Ref. ³⁶]. Based on this model, the nonadiabaticity is determined by the λ_J and λ_{sf} .

We first consider λ_J , which is governed by J . The s - d exchange interaction originates from the interaction between itinerant s electrons and localized d electrons. It is generally independent of the type of sublattices coupling. Hence, we take as representative value $\lambda_J = 1$ nm from the literature [Ref. ³⁶].

In contrast, λ_{sf} depends strongly on the sublattice coupling, as spin-flip process is highly sensitive to the magnetic texture. The staggered antiferromagnetic sublattice induces faster spin

³⁵ Tataru, G. *et al.* Microscopic approach to current-driven domain wall dynamics. *Phys. Rep.* **468**, 213–301 (2008). DOI: <https://doi.org/10.1016/j.physrep.2008.07.003>.

³⁶ Thiaville, A. *et al.* Micromagnetic understanding of current-driven domain wall motion in patterned nanowires. *Europhys. Lett.* **69**, 990–996 (2005). DOI: [10.1209/epl/i2004-10452-6](https://doi.org/10.1209/epl/i2004-10452-6).

dephasing than a ferromagnetic sublattice, resulting in a shorter λ_{sf} . Recent theoretical calculations have revealed a rapidly reduced spin-flip time when transitioning from a ferromagnetic to an antiferromagnetic sublattice [Ref.³⁷]. For a ferromagnetic sublattice, λ_{sf} is generally on the order of several nanometers³⁸, whereas in antiferromagnet it can be as short as 0.5 nm³⁹.

Accordingly, we can adopt $\lambda_{sf} = 5$ nm for the ferromagnetic lattice³⁶ and $\lambda_{sf} = 0.5$ nm for the antiferromagnetic lattice³⁹. Using these values, we estimate $\beta = 0.04$ for the ferromagnetic lattice, and $\beta = 4$ for the antiferromagnetic lattice. The estimated β is in 10^{-2} order for a ferromagnet, consistent with general expectations, whereas in an antiferromagnet it can be up to two orders of magnitude larger, owing to the strong mistracking of electron spins when travelling through antiferromagnetic sublattices.

In response to the reviewer's comment, we modified the **Discussion Section** and added the above discussion on the origin of large nonadiabaticity in NCO in Line # 222–242 in the revised manuscript (text highlighted in blue).

³⁷ Lu, H. *et al.* Magnetic structure-dependent ultrafast spin relaxation in magnet CrI₃: A time-domain ab Initio study. *Nano Letters* **24**, 8940-8947 (2024). DOI: [10.1021/acs.nanolett.4c01809](https://doi.org/10.1021/acs.nanolett.4c01809).

³⁸ Bass, J. *et al.* Spin-diffusion lengths in metals and alloys, and spin-flipping at metal/metal interfaces: an experimentalist's critical review. *J. Phys.: Condens. Matter* **19**, 183201 (2007). DOI [10.1088/0953-8984/19/18/183201](https://doi.org/10.1088/0953-8984/19/18/183201).

³⁹ Zhang, W. *et al.* Spin Hall effects in metallic antiferromagnets. *Phys. Rev. Lett.* **113**, 196602 (2014). DOI: <https://doi.org/10.1103/PhysRevLett.113.196602>.

2. The relativistic effect: Recent paper [Science 370, 1438 (2020)] have demonstrated the relativistic kinematics of domain wall motion at high speed. The authors' observations also show a very high domain wall velocity up to 1 km/s. However, in Figure 3d. it appears that the domain wall velocity diverges at large applied current. What is the origin of this divergence? Is there any relativistic effect in domain wall motion at such high velocity?

Response: We thank the reviewer for this comment. We estimated the theoretical limit from the magnon group velocity v_g^{\max} , using the theory from Refs. ^{40,41}. v_g^{\max} can be derived from the spin-wave dispersion relation, and expressed as $v_g^{\max} = \frac{2A}{dS}$, where A is the exchange stiffness, d the lattice constant and $S = |S_1| + |S_2|$ the total angular momentum.

In the case of NCO, we take $d = 0.8$ nm. We estimate A from the DW width: $\delta = \pi \sqrt{\frac{A}{K_u}}$. Taking $\delta = 39 \pm 13$ nm from NV measurement and $K_u = 0.2$ MJ/m³ [from Ref. ⁴²], we obtain $A = 3.2 \pm 2.2 \times 10^{-11}$ J/m. The magnetic moments in NCO are carried by antiferromagnetically coupled Ni ($1.5 \mu_B/\text{Ni}$) and Co ($3.5 \mu_B/\text{Co}$) sublattice [Fig. 2 of Ref. ⁴³]. Using $g_{\text{Co}} = g_{\text{Ni}} = 2.2$, we obtain $S = \frac{5\mu_B}{\gamma} / \text{f.u.} = 3.62 \times 10^{-6}$ kg/(m · s). Based on these parameters, we estimate the theoretical limit of DW velocity as $v_g^{\max} = 22 \pm 15$ km/s.

Here, a current density of 4×10^{12} A/m² would be required to observe relativistic effect in NCO, according to the linear expansion of Fig. 3e. Moreover, under the relativistic condition, DW velocity gets saturated, which contradicts the divergent trend observed in Fig. 3d. Therefore, we think this velocity divergence does not originate from relativistic effect. Such velocity upturns have

⁴⁰ Shiino, T. *et al.* Antiferromagnetic domain wall motion driven by spin-orbit torques. *Phys. Rev. Lett.* **117**, 087203 (2016). DOI: <https://doi.org/10.1103/PhysRevLett.117.087203>.

⁴¹ Caretta, L. *et al.* Relativistic kinematics of a magnetic soliton. *Science* **370**, 1438-1442 (2020). DOI: [10.1126/science.aba5555](https://doi.org/10.1126/science.aba5555).

⁴² Shen, Y. *et al.* Tuning of ferrimagnetism and perpendicular magnetic anisotropy in NiCo₂O₄ epitaxial films by the cation distribution. *Phys. Rev. B* **101**, 094412 (2020). DOI: <https://doi.org/10.1103/PhysRevB.101.094412>.

⁴³ Xu, X. *et al.* Epitaxial NiCo₂O₄ film as an emergent spintronic material: Magnetism and transport properties. *J. Appl. Phys.* **132**, 020901 (2022). DOI: <https://doi.org/10.1063/5.0095326>.

been previously observed in metallic ferromagnets^{44,45} as well as in insulating ferrimagnet⁴⁶. It is primarily attributed to Joule heating and Oersted field, both of which become more pronounced at high current density regime. The current-induced out-of-plane Oersted field facilitates the DW propagation, while the Joule heating can reduce the uniaxial anisotropy, thereby accelerating magnetization rotation during DW motion.

In response to this comment, we added a new section in the Supplementary Material (**Supplementary Note 6: Theoretical limit of DW velocity in NCO**) including the above discussion of the theoretical velocity limit of DW motion in NCO.

⁴⁴ Yang, S. H. *et al.* Domain-wall velocities of up to 750 m s⁻¹ driven by exchange-coupling torque in synthetic antiferromagnets. *Nat. Nanotechnol.* **10**, 221–226 (2015). DOI : <https://doi.org/10.1038/nnano.2014.324>.

⁴⁵ Ryu, KS. *et al.* Chiral spin torque at magnetic domain walls. *Nature Nanotech* **8**, 527–533 (2013). DOI : <https://doi.org/10.1038/nnano.2013.102>.

⁴⁶ Vélez, S. *et al.* High-speed domain wall racetracks in a magnetic insulator. *Nat Commun* **10**, 4750 (2019). DOI: <https://doi.org/10.1038/s41467-019-12676-7>.

3. Minor comments: There are several minor mistakes and suggestions. (3-1) for Figure 1: On line 97th, Figure 1c presents transmission electron microscopy (TEM) data, but the manuscript explains it as a magnetic hysteresis loop. Please provide an appropriate explanation of the TEM data.

(3-2) for Figure 3: On line 165th, Figure 4e should be corrected to Figure 3e.

(3-3) Suggestion I: In Figure 2b, please provide the scanning line of the NV center measurement results shown in Figures 2c and 2d. This will be helpful for readers.

(3-3) Suggestion II: Please include a linear fitting line in Figure 3e.

End of review report

Response: We thank the reviewer for checking these minor but important points. We have made the following modifications according to the suggestions:

1. We apologize for the mistake, line 97 should be Fig. 1d instead of Fig. 1c. We have corrected this mistake in the revised manuscript.
2. We apologize for the typo. We have corrected Fig. 4e to Fig. 3e in the revised manuscript and we have checked all the manuscript again to avoid any mistake.
3. We have added the dashed lines as is suggested and according to descriptions in the caption.
4. We have added the linear fitting of Fig. 3e, and thanks for the suggestion.

Reviewer #4

(Remarks to the Author):

The authors of this manuscript report fast domain wall (DW) motion at velocity up to 1 km/s in NiCo₂O₄ ferrimagnetic spinel oxide under relatively low current density via spin transfer torque (STT). They attribute this rapid motion to the high spin polarization and large nonadiabaticity arising from the staggered spin configuration. In addition, they report inertial effects enabling DW motion for ~1 ns after turning off the pulse. While the observation of rapid domain wall motion at low current density is indeed advantageous from the viewpoint of racetrack memory, it is hard for me to think that work represent a major technological advance. Fast domain wall motion has already been reported multiple times in the literature, not only in the STT-driven cases as cited by the authors, but also in spin-orbit torque systems. It is not clear what specific advantage arises from achieving domain wall motion purely by STT, and the explanation for the fast motion does not show new physics.

Moreover, significant inertia could in fact be an obstacle for precisely controlling domain wall position using the current pulse. Overall, although the data are interesting, I am not convinced that this work meets the standards required for the publication in Nature Communications. I therefore do not recommend publication.

Response: We thank the reviewer for providing critical comments and questions. We elaborate below on the advances and significance of this work.

There are several reasons to pursue studies of high-velocity DW motion in different material systems and using different driving methods. Advancing racetrack technology is certainly one. However, finding new types of materials that support fast current-driven DW motion is equally relevant to expand the scope of spintronics and improve the fundamental understanding of phenomena leading to high DW mobilities. Moreover, we do not argue that STT-driven DW motion should be pitched against SOT, as both methods have their own advantages and disadvantages. The two methods allow for driving DW motion in different material systems using different geometries and are not mutually exclusive. Therefore, we believe that our investigation of DWs in a spinel oxide and the demonstration of extremely high STT-DW mobility and nonadiabaticity in this material system are relevant for a broad audience.

Regarding the comparison of SOT vs STT: The reviewer noted that SOT allows for very fast DW motion, as reported in ferrimagnets such as GdCo⁴⁷ and Bi-YIG⁴⁸. However, both systems require an in-plane bias magnetic field, e.g., 1.4 kOe for the case of GdCo, which is unfavorable for device integration. Moreover, the required current density and bias magnetic field for reaching DW velocity of 4 km/s in Bi-YIG are 3×10^{12} A/m² and 280 Oe, respectively. In this work, we demonstrate that NCO exhibits a DW velocity exceeding 1 km/s under a current density of just 2×10^{11} A/m² at room temperature, without any external magnetic field. An important feature of SOT is that it can drive DW motion in an insulating magnetic layer by passing a current in an adjacent heavy metal layer. Conversely, STT can drive DW motion in a single-layer magnetic structure, eliminating the need for a heavy-metal layer as a spin source. In our case, only Ni and Co elements are required, without heavy or rare-earth metals. The two methods are therefore complementary. Although SOT has attracted considerable attention for its higher spin-torque efficiency, our results demonstrate that STT can achieve comparable performance and, in certain cases, even surpass SOT. We believe these findings represent an important step toward the realization of efficient DW motion in different material systems and geometries, in addition to high-performance racetrack memory devices.

Regarding the physics, many theoretical studies have highlighted the critical role of nonadiabatic torque in DW dynamics within antiferromagnetically coupled sublattices. However, experimental studies of nonadiabaticity in ferrimagnets remain scarce (to our knowledge, only Ref. ⁴⁹ reports such an investigation). In this work, we systematically studied DW dynamics and clarified the role of nonadiabatic torque in driving DW motion in the ferrimagnetic system with staggered magnetic moments. It offers substantial new insights for the spintronics community.

Regarding DW inertia, we report for the first time its observation in ferrimagnets. As the reviewer noted, inertia can hinder precise control of DW position. This was also the reason for us to investigate DW inertia in detail in ferrimagnetic NCO. We should emphasize that the extracted relaxation time $\tau \lesssim 1$ ns is significantly shorter than values reported for ferromagnets, such as

⁴⁷ Cai, K. *et al.* Ultrafast and energy-efficient spin-orbit torque switching in compensated ferrimagnets. *Nat Electron* **3**, 37–42 (2020). DOI: <https://doi.org/10.1038/s41928-019-0345-8>.

⁴⁸ Caretta, L. *et al.*, Relativistic kinematics of a magnetic soliton. *Science* **370**, 1438-1442 (2020). DOI: [10.1126/science.aba5555](https://doi.org/10.1126/science.aba5555).

⁴⁹ Okuno, T. *et al.* Spin-transfer torques for domain wall motion in antiferromagnetically coupled ferrimagnets. *Nat Electron* **2**, 389–393 (2019). DOI: <https://doi.org/10.1038/s41928-019-0303-5>.

permalloy (11.5 ns) and $\text{W/Co}_{20}\text{Fe}_{60}\text{B}_{20}$ (2–5.5 ns). This indicates the ferrimagnet is more immune to inertia than the ferromagnetic DWs. It provides important information for researchers who are developing ferrimagnetic spintronic devices.

In conclusion, our manuscript reports the fast DW motion in the ferrimagnetic spinel oxide with low magnetization, high spin polarization, and large nonadiabaticity, as well as a systematic investigation of nonadiabatic STT and ferrimagnetic DW inertia. We believe it offers both fundamental insights and practical guidance for engineering fast DW motion in ferrimagnetic systems.

Below I provide several minor comments:

1. The manuscript claims that the characteristic time of ~ 1 ns is shorter than in typical ferromagnets, but a massless domain wall motion has previously been reported [J. Vogel Phys. Rev. Lett. 108, 247202 (2012)].

Response: We appreciate the reviewer's valuable comment. We carefully reviewed the reference⁵⁰ [J. Vogel Phys. Rev. Lett. 108, 247202 (2012)]. In this work, the authors measured DW inertia by fitting the delay in DW motion corresponding to its acceleration at the onset of a current pulse. They reported delay times of 4.4 ± 2.9 ns under a current density of 7.7×10^{11} A/m², and 4.8 ± 5 ns under 7.7×10^{12} A/m². The analysis assumes that the DW remains immobile during the current-pulse ramp, which has a rise time of 4 ns. Consequently, the estimated acceleration times are 0.4 ± 2.9 ns and 0.8 ± 5 ns for these two current densities, respectively. The authors thus concluded that the DW inertia is negligible and DW mass is very small in the Pt/Co/AlO_x film. They attributed these to the high damping ($\alpha = 0.5$). However, it should be noted that the relatively long pulse rise time (4 ns) introduces significant uncertainty, resulting in large error bars for the extracted acceleration times. Furthermore, spin-orbit torque was not considered in their analysis of DW acceleration, as expressed $\tau_A = \frac{1+\alpha^2}{\gamma|\alpha H_K + \frac{\pi}{2}H_{SH}|}$ (Eq. (2) of Ref.⁵¹).

Comprehensive studies of ferromagnetic DW inertia have been conducted for STT-driven DW motion in permalloy⁵² and SOT-driven DW motion in CoFeB⁵¹. The reported characteristic times are 11.5 ns and 2–5.5 ns, respectively. We thus conclude that the extracted $\tau \lesssim 1$ ns in NCO is notably shorter than that observed in ferromagnets. This is the first observation of DW inertia in ferrimagnets. We believe the demonstration of small characteristic time provides important information for the researchers working on ferrimagnetic spintronics.

⁵⁰ Vogel, J. *et al.* Direct observation of massless domain wall dynamics in nanostripes with perpendicular magnetic anisotropy. *Phys. Rev. Lett.* **108**, 247202 (2012). DOI:<https://doi.org/10.1103/PhysRevLett.108.247202>.

⁵¹ Torrejon, J. *et al.* Tunable inertia of chiral magnetic domain walls. *Nat Commun* **7**, 13533 (2016). DOI:<https://doi.org/10.1038/ncomms13533>.

⁵² Thomas, L. *et al.* Dynamics of magnetic domain walls under their own inertia. *Science* **330**, 1810-1813 (2010). DOI:[10.1126/science.1197468](https://doi.org/10.1126/science.1197468).

2. The material used here appears to have high resistivity, which is not favorable in terms of power-consumption. The authors suggested volume-normalized energy consumption, but given the relatively large film thickness used in this study, I question whether this is an appropriate parameter. A comparison in terms of energy per device length might be more relevant.

Response: We thank the reviewer for this interesting question, which we address below.

The NCO used in our manuscript has a resistivity of $800 \mu\Omega \cdot cm$, which is indeed higher than that of typical metallic thin films ($\sim 10^1 \mu\Omega \cdot cm$) and alloys ($\sim 10^2 \mu\Omega \cdot cm$). Nevertheless, we explicitly included the resistivity parameter in our evaluation of energy consumption and found that NCO exhibits extremely low energy consumption due to high mobility of DW even with relatively high resistivity.

To keep into account the resistivity of racetrack devices, Kumar et al⁵³ proposed evaluating the energy consumption using the energy required for $1 \mu m$ DW displacement, given by:

$$\xi = \frac{I^2 R t}{s}, \quad (R1)$$

where I is the injected current, R is the device resistance, t is the pulse duration, and s is the DW displacement. This equation can be rewritten as:

$$\xi = \frac{j^2 \rho \mathcal{A} L}{v_{\text{avg}}}, \quad (R2)$$

where ρ is the resistivity, \mathcal{A} is the cross-sectional area, and L is the racetrack length. Clearly, ξ depends on the device geometry, which makes it challenging to compare the STT efficiency for driving DWs in different material systems (Fig. R2a). To solve this issue, we normalize ξ by the device volume. Accordingly, the volume-normalized ξ_V is defined as:

$$\xi_V = \frac{j^2 \rho}{v_{\text{avg}}}. \quad (R3)$$

With this definition, ξ_V reflects the effective energy consumption per $1 \mu m$ DW displacement. It is proportional to the j , the mobility (j/v_{avg}), and ρ .

⁵³ Kumar, D. *et. al* Ultralow energy domain wall device for spin-based neuromorphic computing. *ACS Nano* **17** (7), 6261-6274 (2023). DOI: [10.1021/acsnano.2c09744](https://doi.org/10.1021/acsnano.2c09744).

In Fig. R2b-d, we compare different normalization factors for evaluating DW device performance, including the suggestion from the referee. Among them, only the volume-normalized energy consumption $\xi_V = \frac{j^2 \rho}{v_{\text{avg}}}$, shown in Fig. R2b, captures the intrinsic material properties independently of the device geometry. The energy consumption per device length, $\xi_L = \frac{j^2 \rho \mathcal{A}}{v_{\text{avg}}}$, shown in Fig. R2c, describes the energy required for displacing a DW associated to the cross-section of the material. The energy consumption per device cross section, $\xi_{\mathcal{A}} = \frac{j^2 \rho L}{v_{\text{avg}}}$, shown in Fig. R2d, describes the energy required for displacing a DW across a device of length L , independently of its cross section. In all cases, we observe that NCO performs well relative to other materials. However, in Fig. R2b, 10-nm and 30-nm-thick NCO devices exhibit similar energy consumption, confirming that the influence of film thickness is effectively removed. Moreover, the enhanced DW performance of antiferromagnets relative to ferromagnets becomes unambiguous in Fig. R2b. Thus, the volume-normalized energy consumption serves as a robust guideline for identifying material properties and systems that enable high STT-driven DW mobility with minimal energy cost.

Fig. R2. Energy consumption for 1 μm DW displacement for different normalization factors. **a**, Energy consumption without normalization **b**, Energy consumption per device volume, same as Fig. 5b in the main text. **c**, Energy consumption per device length. **d**, Energy consumption per device cross-sectional area.

In response to this comment, we added a new section in the Supplementary Material (**Supplementary Note 11: Energy consumption for DW racetrack devices**) including the above discussion and Fig. R2, and referred to this point in the main manuscript.

3. The domain walls appear to be quite wide (tens of nanometers), which does not seem advantageous in terms of domain density (bit-density).

Response: We thank the reviewer for the comment. In the experiment, we measured the DW width of $\delta = 39 \pm 13$ nm for NCO using NV magnetometry. This is comparable to typical width of a ferromagnetic DW, as shown in the following Table. We would like to clarify the distinct difference between the physical DW width δ and DW width parameter Δ , which are related by $\delta = \pi\Delta$. The physical DW width δ is defined as the distance for a 180° spatial rotation of magnetization ($+\mathbf{M} \rightarrow -\mathbf{M}$). In contrast, Δ is a mathematical parameter, appearing in the analytical solution for the magnetization angle profile, $\theta(x) = 2\arctan\left[\exp\left(\frac{x}{\Delta}\right)\right]$. NV magnetometry provides a direct measurement of δ , rather than Δ .

In response to the reviewer's comment, we added the following Table in Supplementary Section 2 to compare typical DW width δ for various ferromagnets, ferrimagnets, and antiferromagnets in the revised manuscript.

Materials	Magnetism	DW width δ (nm)	DW type	Method	References
Ta/CoFeB(1)/MgO	FM	20	Bloch	NV center	[1]
Pt(3)/Co(0.6)/AlO _x (2)	FM	6	Néel	NV center	[1]
Fe	FM	64	/	Calculation	[2]
Co	FM	24	/	Calculation	[2]
NiCo ₂ O ₄	FI	39 ± 13	Bloch	NV center	This work
Mn ₄ N	FI	25	/	Calculation	[3]
Pt/GdCo/TaO _x	FI	35	/	Calculation	[4]
Pt/CoTb/SiN _x	FI	31	/	Calculation	[5]
Bi-YIG	FI	44	/	Calculation	[6]
TmIG	FI	27 ± 6	Néel	NV center	[7]
Mn ₃ Sn	AFM	40	Néel	NV center	[8]
Cr ₂ O ₃	AFM	42 - 65	Mixed Bloch-Néel	NV center	[9]
Mn ₂ Au	AFM	30	/	Calculation	[10]

Table S2 Summary of DW widths δ for various ferromagnets, ferrimagnets, and antiferromagnets. The DW width is extracted by $\delta = \pi\sqrt{\frac{A}{K_u}}$, where A and K_u are exchange stiffness and uniaxial anisotropy constant, respectively.

4. Appropriate citations are needed for the experiments on the in-plane magnetic field dependence of domain wall velocity.

Response: We thank the reviewer for the comment. We carefully investigated and cited following literature in Refs. [4, 6, 11-16] of the revised supplementary material about the in-plane magnetic field dependence of DW velocity:

[4] Caretta, L., Mann, M., Büttner, F., Ueda, K., Pfau, B., Günther, C. M., Helsing, P., Churikova, A., Klose, C., Schneider, M., Engel, D., Marcus, C., Bono, D., Bagschik, K., Eisebitt, S. & Beach, G. S. D. Fast current-driven domain walls and small skyrmions in a compensated ferrimagnet. *Nat. Nanotechnol.* **13**, 1154–1160 (2018).

[6] Caretta, L., Oh, S. H., Fakhru, T., Lee, D. K., Lee, B. H., Kim, S. K., Ross, C. A., Lee, K. J. & Beach, G. S. D. Relativistic kinematics of a magnetic soliton. *Science* **370**, 1438–1442 (2020).

[11] Ryu, K. S., Thomas, L., Yang, S. H. & Parkin, S. S. P. Chiral spin torque at magnetic domain walls. *Nat. Nanotechnol.* **8**, 527–533 (2013).

[12] Filippou, P. C., Jeong, J., Ferrante, Y., Yang, S. H., Topuria, T., Samant, M. G. & Parkin, S. S. P. Chiral domain wall motion in unit-cell thick perpendicularly magnetized Heusler films prepared by chemical templating. *Nat. Commun.* **9**, 4653 (2018).

[13] Emori, S., Bauer, U., Ahn, S. M., Martinez, E. & Beach, G. S. D. Current-driven dynamics of chiral ferromagnetic domain walls. *Nat. Mater.* **12**, 611–616 (2013).

[14] Avci, C. O., Rosenberg, E., Caretta, L., Büttner, F., Mann, M., Marcus, C., Bono, D., Ross, C. & Beach, G. S. (2019). Interface-driven chiral magnetism and current-driven domain walls in insulating magnetic garnets. *Nat. Nanotech.* **14**, 561–566 (2019).

[15] Yang, S. H., Ryu, K. S. & Parkin, S. S. P. Domain-wall velocities of up to 750 m s⁻¹ driven by exchange-coupling torque in synthetic antiferromagnets. *Nat. Nanotechnol.* **10**, 221–226 (2015).

[16] Kato, N., Kawaguchi, M., Lau, Y. C., Kikuchi, T., Nakatani, Y., & Hayashi, M. Current-induced modulation of the interfacial Dzyaloshinskii-Moriya interaction. *Phys. Rev. Lett.* **122**, 257205 (2019).

5. The fact that the domain wall moves along the current direction by STT is intriguing but is treated rather superficially. The authors should discuss whether previous studies on the directionality of STT exist, and whether alternative explanations beyond STT are possible for the opposite STT.

Response: We thank the reviewer for the insightful comment. Other studies have reported that DW can move in the current direction when driven by STT, such as Ref.⁵⁴ and Ref.⁵⁵. This behavior has been attributed to a negative spin polarization of conduction electrons.

Let us elaborate on the physical mechanism. As illustrated in the schematic figure R3, consider an initial Up | Down DW configuration. When a current is applied from right to left, conduction electrons flow oppositely (left to right) and become spin-polarized by the localized magnetization. If the magnetic film exhibits positive spin polarization (second panel case), the resulting spin-polarized current points upward. The transfer of angular momentum to the localized magnetic moments results in the DW motion toward the right—opposite to the current direction. Conversely, if the magnetic material has a negative spin polarization, the spin-polarized current drives the DW toward the left, resulting in DW motion in the same direction as the applied current (third panel).

Fig. R3. Schematic illustration of DW motion direction with respect to the injected current.

⁵⁴ Filippou, P.C. *et al.* Chiral domain wall motion in unit-cell thick perpendicularly magnetized Heusler films prepared by chemical templating. *Nat Commun* **9**, 4653 (2018). DOI: <https://doi.org/10.1038/s41467-018-07091-3>.

⁵⁵ Ghosh, S. *et al.* Current-driven domain wall dynamics in ferrimagnetic nickel-doped Mn₄N films: very large domain wall velocities and reversal of motion direction across the magnetic compensation point. *Nano Letters* **21** (6), 2580-2587 (2021). DOI: [10.1021/acs.nanolett.1c00125](https://doi.org/10.1021/acs.nanolett.1c00125).

In the case of NCO, the magnetic moments are carried by eight Ni ions ($1.5 \mu_B/\text{Ni}$) on octahedral sites and eight Co ions ($3.5 \mu_B/\text{Co}$) on tetrahedral sites, which are antiferromagnetically aligned. The resultant ferrimagnetism is determined by the Co sublattice. However, the electronic spin states at the Fermi level are dominated by Ni, leading to a strong negative spin polarization. Theoretical calculations predict a negative spin polarization, which has experimentally been confirmed through magnetic tunnel junction measurements, yielding $P = -0.73$. Consequently, DW is expected to move in the same direction as the injected current (third panel), in agreement with our experimental observations.

In response to the reviewer's comment, we added a new note to the revised manuscript and this discussion in Supplementary Material (**Supplementary Note 3: Discussion on DW motion direction**).

6. The resistance of the sample should be specified. Given the high resistivity of the material, the resistance is likely much larger than 50Ω , raising concerns about impedance mismatch when injecting 1 ns current pulses. The inset in Fig. 4b shows a current pulse profile; it should be clarified whether this waveform was measured after transmission through the sample using an oscilloscope, or whether it simply obtained without the sample.

Response: The two-terminal resistance of our device is about $1.5 - 2.6 \text{ k}\Omega$ for 30 and 10 μc NCO devices, which includes the racetrack and the contact resistance. For the inset of Fig. 4b, we presented the pulser waveform measured without sample connected. To investigate the influence of impedance mismatch on the pulse waveform. We compared the pulse shapes measured without and with sample configurations (Fig. R4a). Figures R4b and c show the results for 1-ns and 3-ns pulses. These results clearly indicate that impedance mismatch does not distort pulse shape during the transmission through the sample.

However, the magnitude of pulse must be carefully calibrated because impedance mismatch causes pulse reflection. The reflection can be quantified with the coefficient:

$$\Gamma = \frac{R_{\text{load}} - Z_0}{R_{\text{load}} + Z_0},$$

where R_{load} is the sample resistance and $Z_0 = 50 \Omega$ is the pulser impedance. According to this equation, if $R_{\text{load}} = 50 \Omega$, $\Gamma = 0$, which means a perfect match without reflection. In this matched case, the voltage on sample equals the set output voltage: $V_{\text{load}} = V_{\text{set}}$. When $R_{\text{load}} \neq 50 \Omega$ ($\Gamma \neq 0$), V_{load} is not equal to V_{set} anymore and must be recalculated as,

$$V_{\text{load}} = V_{\text{set}}(1 + \Gamma) = V_{\text{set}} \frac{2R_{\text{load}}}{R_{\text{load}} + Z_0}.$$

Figure R4d plots the V_{load} as a function of R_{load} . As expected, $V_{\text{load}} = V_{\text{set}}$ when $R_{\text{load}} = 50 \Omega$, and V_{load} approaches $2V_{\text{set}}$ if $R_{\text{load}} \gg 50 \Omega$. In our experiment, we employed this method to determine V_{load} for each device to ensure the accurate calculation of current density.

Fig. R4 a, Measurement setups for the circuits without and with samples. **b, c**, Shape comparisons of 1-ns and 3-ns pulses under and without sample configurations. **d**, The plot of $\frac{V_{\text{load}}}{V_{\text{set}}}$ as a function of the R_{load} .

In response to the reviewer's comment, we have made the following modifications in the revised manuscript:

1. We replaced the inset of Fig. 4 with the waveform measured after transmission.
2. We added the discussion on the impedance mismatch in **Supplementary Note 9**.

Summary of changes

All changes to the text have been highlighted in blue in the revised manuscript and supplementary information files. Additionally, we have included the following changes:

Revised manuscript:

1. In lines #35-45 of **Introduction Section**, we explained the more detailed motivation of studying STT-driven DW motion in NCO.
2. In line # 102, we modified the mistake of Fig. 1d, rather than Fig. 1c.
3. In line # 144, we added: “(see Supplementary Note 3 for more detail)”.
4. In line # 167, we added: “(see Supplementary Note 6 for the discussion on theoretical velocity limit)”.
5. In line # 172, we corrected the typo of Fig. 4e to Fig. 3e and added: “(see Supplementary Note 7 for DW dynamics formulation for NCO)”.
6. In lines #222-242 of **Discussion Section**, we discussed the mechanism of large nonadiabaticity in NCO.
7. We modified Fig. 2b to show the region of VN line scans, with the caption: “The red and blue dashed lines indicate the regions where line scans were performed for figures c and d, respectively”.
8. We added the linear fitting in Fig. 3e, with the caption: “The black line is the linear fit from which DW mobility can be extracted”.
9. In Fig. 4b, we replaced the inset figure with the pulse shapes transmitted after device and added the description in the caption.
10. In Fig. 4a and 4b, we deleted the improper describing words “simulation” and “experiment”.

Revised Supplementary material

1. In Supplementary Note 1, we added “Table S2 Summary of DW widths δ for various ferromagnets^{1,2}, ferrimagnets³⁻⁷, and antiferromagnets⁸⁻¹⁰”, with a description in line #32: “This is comparable to the typical width of a DW defined by $\delta = \pi \sqrt{\frac{A}{K_u}}$, where A and K_u are exchange stiffness and uniaxial anisotropy constant, respectively, as shown in Supplementary Table 2”.
2. In Supplementary Note 2, we added references [4,6, 11-16] in line # 54 and # 64 about in-plane field dependence of DW velocity.

3. We added a new note: Supplementary Note 3: Discussion on DW motion direction.
4. We added a new note: Supplementary Note 6: Theoretical limit of DW velocity in NCO.
5. We added a new note: Supplementary Note 7: DW dynamics formulation for ferrimagnetic NCO.
6. In line #232, we corrected the typo: “precessional”.
7. We added a new note: Supplementary Note 9: Discussion on the influence of impedance mismatch.
8. In Supplementary Note 11, we added the discussion in line # 317-329 on the comparison of different strategies for normalizing energy consumption and the validity of volume-normalized energy consumption.

In addition, we have renumbered all figures and equations accordingly, and have rechecked all the text to avoid any mistake in the revised manuscript and supplementary manuscript.

Reviewer #1

(Remarks to the Author)

My main concern is the validity of the theoretical model. The authors justified that they use the 1D model with antiferromagnetically coupled sublattices, which is an unsatisfied answer. The same model has been extensively used in the previous studies of ferrimagnets such as CoTb, GdFeCo, or antiferromagnets. Therefore, nothing is new if the authors rely on such simplified model to explain the experimental results, and there is no reason to believe NCO has new physics compared to other ferrimagnets or antiferromagnets. If the authors believe there is new physics in NCO, I suggest the author to seriously consider more dedicated theories that can capture the unique characteristics of NCO, such as the complicated lattice structure and exchange interactions beyond the nearest neighbor coupling. At this stage, I do not recommend its publication in nature communications.

Response: We thank the reviewer for taking additional time to review our manuscript.

We understand the reasoning by the referee: if a well-established model is used to interpret the data, then nothing new emerges from the data. However, we do not agree with the implicit logic of this comment. Established models do not prevent new discoveries. Rather, they provide a benchmark that makes new discoveries visible. Novelty not only arises from model violations, but also from revealing parameters whose values become the discovery, such as the extremely high domain wall (DW) mobility and nonadiabaticity in our case, and from extending models to different materials classes, such as spinel oxides rather than ferromagnetic metals or ferrimagnetic alloys.

In fact, a majority of novel findings reported across different fields make use of established models for their interpretation. Famous examples include the “discovery” of massless electron dispersions in a variety of 2D materials, which follows a rather simple tight-binding model first established in the 1940s. The 1D model is widely used to formulate DW dynamics in racetracks under various driving forces (magnetic field, STT or SOT) across a broad range of materials. Although different material systems employ the same 1D model, this does not imply a lack of new physics. For example, the 1D model was applied to CoTb to interpret the increase of DW velocity

near the compensation point¹, to GdFeCo to support evidence for STT-driven DW motion in a ferrimagnetic alloy², to Bi-YIG to explain the saturation of DW velocity³, and to Fe₃GeTe₂ and Fe₃GaTe₂ to report current-induced motion in 2D materials^{4,5}.

In regard to this manuscript, we report an extremely large STT-driven DW velocity exceeding 1 km/s under a moderate current density of 2×10^{11} A m⁻² in a single-layer spinel oxide material without the need for an external magnetic field. The novelty lies in the following experimental observations:

- 1) First demonstration of current-driven DW motion in the ferrimagnetic spinel oxides;
- 2) Demonstration of the giant nonadiabatic character of the STT and its role in high DW mobility in this class of materials;
- 3) First observation of DW inertia in a ferrimagnet with a characteristic time ~ 1 ns, smaller than typical ferromagnets;
- 4) Lowest depinning threshold, high DW mobility and lowest energy consumption for DWs driven by STT compared to most ferromagnets and ferrimagnets.

We appreciate the reviewer's thoughtful suggestion to consider a more dedicated theory. A complex and more complete theory may reveal interesting physics, but goes beyond the scope of this paper and our own capabilities. Regarding the exchange interactions, the nearest-neighbor coupling is typically much stronger than the next-nearest-neighbor coupling, and is therefore widely considered as the dominant contributor to magnetic order. This is also applicable to NiCo₂O₄ (NCO) which exhibits an inverse spinel structure. The nearest-neighbor exchange coupling J_{A-B} , originating from Ni ions on octahedral sites (A sites) and Co ions on tetrahedral sites (B sites), is

¹ Siddiqui, S. *et al.*, Current-induced domain wall motion in a compensated ferrimagnet. *Phys. Rev. Lett.* **121**, 057701 (2018). DOI: <https://doi.org/10.1103/PhysRevLett.121.057701>.

² Okuno, T. *et al.* Spin-transfer torques for domain wall motion in antiferromagnetically coupled ferrimagnets. *Nat Electron* **2**, 389–393 (2019). DOI: <https://doi.org/10.1038/s41928-019-0303-5>.

³ Caretta, L. *et al.*, Relativistic kinematics of a magnetic soliton. *Science* **370**, 1438-1442 (2020). DOI: [10.1126/science.aba5555](https://doi.org/10.1126/science.aba5555).

⁴ Zhang, W., Ma, T., Hazra, B.K. *et al.* Current-induced domain wall motion in a van der Waals ferromagnet Fe₃GeTe₂. *Nat Commun* **15**, 4851 (2024). DOI : <https://doi.org/10.1038/s41467-024-48893-y>.

⁵ Guan, Y., Wu, Y., Zhang, Y. *et al.* Highly efficient current-induced domain wall motion in a room temperature van der Waals magnet. *Nat Commun* **16**, 10790 (2025). DOI: <https://doi.org/10.1038/s41467-025-66637-4>.

larger than the next-nearest-neighbor couplings (J_{A-A} and J_{B-B})⁶, resulting in ferrimagnetism⁷. Moreover, considering a complicated lattice structure or including exchange interactions beyond the nearest neighbors may modify the exchange energy by $E_{\text{ex}} = (J_{A-B} + J_{A-A} + J_{B-B}) \sum_{ij} \mathbf{m}_i \cdot \mathbf{m}_j$, and consequently influences the effective field term ($\mathbf{H}_{\text{eff}} = -\frac{1}{M_s} \frac{\partial E}{\partial \mathbf{m}}$) of the LLG equation. This influence appears, for example, in the DW width $\delta = \pi \sqrt{\frac{A}{K_u}}$. However, it does not affect the analytical expression of 1D model, as an effective exchange coupling $J_{\text{eff}} = J_{A-B} + J_{A-A} + J_{B-B}$ can always be introduced to account for the modification from the exchange interactions.

In summary, our experimental results demonstrate that spinel oxide ferrimagnets represent a novel material platform for STT-based DW devices, offering unexpected and exceptionally high DW mobility combined with low pinning and minimal energy consumption. Using NV microscopy, we provided the first insights into the type and width of DWs in NCO. By incorporating the 1D model, we further revealed the giant nonadiabatic STT and the short timescale of DW inertia in this ferrimagnetic compound. We therefore believe that our findings provide meaningful advances toward the understanding and development of efficient current-driven DW motion in different materials and racetrack systems.

⁶ Srivastava, C. M., Srinivasan, G., & Nanadikar, N. G. Exchange constants in spinel ferrites. *Phys. Rev B*, **19**, 499 (1979). DOI: <https://doi.org/10.1103/PhysRevB.19.499>.

⁷ Xu, X *et al.* Epitaxial NiCo₂O₄ film as an emergent spintronic material: Magnetism and transport properties. *J. Appl. Phys.* **132**, 020901 (2022). DOI: <https://doi.org/10.1063/5.0095326>.

Reviewer #2

(Remarks to the Author)

Key point of the submitted manuscript is the exceptionally high domain wall mobility driven by spin-transfer torque (STT). My previous report was overall positive, and the authors' replies to my remarks as well as the corresponding changes in the revised manuscript are satisfactory. After reviewing the other referees' reports, I noted that Referee 4 raised serious concerns regarding the novelty of the work, claiming that high domain wall mobilities have been reported before — however, without providing any specific evidence. Referee 1 (in remark #4) compares the present results seriously to a previous study that demonstrated similarly high domain wall mobilities, but in a very different system and driven by spin-orbit torque (SOT), a distinct mechanism. I find the authors' response to this remark convincing. In conclusion, I recommend publication of the manuscript in Nature Communications.

Response: We thank the reviewer for evaluating the revised version of our manuscript, including arguments presented in the rebuttal letter, and for recommending publication.

Reviewer #3

(Remarks to the Author)

I would like to express my gratitude to the authors for their tremendous efforts in alleviating my concerns. Therefore, I am pleased to recommend the publication of this manuscript.

Response: We thank the reviewer for the clear appreciation of our work and for recommending publication of this manuscript.

Reviewer #4

(Remarks to the Author):

The authors have responded well to the reviewers' comments and have provided persuadable answers in several aspects. While I believe that the NCO material presents meaningful results in terms of the development of domain-wall-motion-based devices, I still question whether this is a significant advance. Although spin-transfer torque (STT) in ferrimagnets has not been extensively studied, I do not think the results present a significant advance compared to previous ferrimagnet studies, which required an external magnetic field to achieve fast domain-wall motion. The need for an external field is not a major issue, as it can be easily expected if the Dzyaloshinskii–Moriya interaction (DMI) is tuned to stabilize a Néel wall.

While the introduction of a new material platform is certainly of interest and could stimulate further studies, however, because the current findings can still be explained within the scope of existing models, it is hard for me to agree that this work 'improve the fundamental understanding of phenomena leading to high DW mobilities.'

By the way, I think presenting the longitudinal field H_x dependence would constitute a genuine new finding. Since non-adiabaticity is inversely proportional to the domain-wall width (in terms of mistracking), verifying whether the domain-wall mobility decreases as H_x modifies the wall width would significantly enhance the novelty of the work.

Response: We appreciate the reviewer's positive remarks on the last-round response. We further elaborate below on the significance of this work in order to address the reviewer's concerns. We also performed additional analysis of the DW velocity as a function of the applied magnetic field.

The concern that fast field-free STT-driven DW motion may not represent a significant advance compared to in-field SOT-driven DW motion is debatable. In fact, a multitude of studies have been dedicated to developing field-free SOT switching precisely because in-field operation is not always possible or practical in integrated memories. Additionally, we believe that STT-driven DW motion does not compete with SOT, but rather complements it, as these two distinct mechanisms have different advantages and disadvantages. Currently, the main drawbacks of STT compared with SOT are its lower efficiency and slower operation speed. Recent work has identified

the ferrimagnetic Heusler alloy Mn_3Ge as a promising material system, in which STT-MTJs⁸ can achieve performance comparable to SOT-MTJs in terms of switching speed, reliability, and energy consumption. Our work demonstrates that STT can likewise achieve comparable performance to SOT-driven DW motion in a single-layer ferrimagnet like NCO, owing to its large nonadiabaticity and high spin polarization. These advances will motivate researchers to explore material properties and systems with even better STT performance in the future.

As argued in response to Referee 1, interpreting results using an existing model does not imply a lack of novelty. Applying this statement retroactively would wipe off novelty from the large number of experimental studies of DWs performed in recent years, which use this model for analysis and data simulations. Besides the demonstration of STT-driven DWs in a new class of materials, we believe that the novelty of our study lies in the exceptionally high nonadiabaticity, STT mobility, and inertia characteristic of this type of ferrimagnetic system. These findings are new and cannot be derived a priori from the 1D model.

As the reviewer noted, STT in ferrimagnets has rarely been experimentally examined, despite extensive theoretical works suggesting that it dominates DW motion in ferrimagnets and antiferromagnets with staggered magnetic moments^{9,10,11,12,13}. The 1D model helps us parametrize the origin of the exceptional STT performance in NCO, providing a useful phenomenological framework to optimize and compare materials enabling high DW mobility, but does not predict which material performs better than others or which material exhibits giant nonadiabatic STT, or high spin polarization combined with low magnetization.

⁸ Garg, C., Filippou, P.C., Ikhtiar *et al.* Ferrimagnetic Heusler tunnel junctions with fast spin-transfer torque switching enabled by low magnetization. *Nat. Nanotechnol.* **20**, 360–365 (2025). <https://doi.org/10.1038/s41565-024-01827-7>.

⁹ Nakane, J. J. & Kohno, H. Microscopic calculation of spin torques in textured antiferromagnets. *Phys. Rev. B* **103**, L180405 (2021). DOI: <https://doi.org/10.1103/PhysRevB.103.L180405>.

¹⁰ Park, H.-J. *et al.* Numerical computation of spin-transfer torques for antiferromagnetic domain walls. *Phys. Rev. B* **101**, 144431 (2020). DOI: <https://doi.org/10.1103/PhysRevB.101.144431>.

¹¹ Jing, K. Y., Sun, Z. Z., & Wang, X. R. Current-driven domain wall motion in ferrimagnetic nanowires. *Phys. Rev. B*, **110**, 054414 (2024). DOI: <https://doi.org/10.1103/PhysRevB.110.054414>.

¹² Hals, K. M., Tserkovnyak, Y., & Brataas, A. Phenomenology of current-induced dynamics in antiferromagnets. *Phys. Rev. Lett.* **106**, 107206 (2011). DOI: <https://doi.org/10.1103/PhysRevLett.106.107206>.

¹³ Swaving, A. C., & Duine, R. A. (2011). Current-induced torques in continuous antiferromagnetic textures. *Phys. Rev. B* **83**, 054428 (2011). DOI: <https://doi.org/10.1103/PhysRevB.83.054428>.

Lastly, we appreciate the reviewer's thoughtful suggestion to consider field-induced modifications of the DW width and its effect on the DW mobility. We present the in-plane field dependence of DW velocity below and in the revised Fig. S2 of Supplementary Note 2.

Fig. S2 DW velocities as functions of in-plane magnetic fields. a, b, H_x and H_y dependence of DW velocities for Down | Up and Up | Down domain configurations. The current density and pulse duration are set to 8.1×10^{10} A m⁻² and 2 ns, respectively. The error bars are the standard deviations of the fits. c, d, Calculated DW width change Δ/Δ_0 as a function of H_x and H_y .

According to the NV magnetometry scans, the DWs in NCO are of Bloch type, with magnetization along y (i.e., perpendicular to the current direction x). Therefore, the DW width Δ is mostly sensitive to a field applied along y rather than x . In the revised Supplementary Information, we theoretically calculate the influence of an in-plane field H_P on Δ by minimizing the DW energy based on Ref. 14 (again the 1D model, sorry!). We obtain

$$\Delta = \Delta_0 \left[1 - \frac{H_K}{2H_U} \cos^2 \psi + \frac{\pi H_P}{2H_U} \cos(\psi - \psi_H) \right] \quad (R1)$$

where $\Delta_0 = \sqrt{\frac{A}{K_U}}$, ψ and ψ_H are the angle of wall magnetization and applied field relative to the x -direction, respectively, $\mu_0 H_U = \frac{2K_U}{M_s}$ is the effective uniaxial anisotropy field, and $\mu_0 H_K = \frac{\ln(2)}{\delta} t M_s = 80 \pm 20$ mT is the demagnetizing field due to magnetic charges on the walls, which leads to a preferred Bloch-wall chirality. We estimate $\mu_0 H_U = 2.8$ T by taking $K_U = 0.2$ MJ/m³ and $M_s = 150$ kA/m, which is significantly larger than the demagnetizing field $\mu_0 H_K$ (~ 80 mT) and the maximum applied field $\mu_0 H_P$ (≤ 40 mT).

Therefore, if a magnetic field $H_P = H_x$ (Figs. S2a and 2c) is applied perpendicular to the wall magnetization, we have

$$\Delta \approx \Delta_0 \left[1 + \frac{\pi^2}{8H_U H_K} H_x^2 \right], \quad (R2)$$

whereas if $H_P = H_y$ (Figs. S2b and 2d) we get

$$\Delta \approx \Delta_0 \left[1 + \frac{\pi}{2H_U} H_y \right]. \quad (R3)$$

Figures S2c and 2d show the calculated Δ/Δ_0 as a function of the applied magnetic fields H_x and H_y . The behavior of Δ/Δ_0 differs under these two field directions. For H_x , the change in DW width depends on the anisotropy field H_K , because H_x tilts the wall magnetization and thereby modifies the demagnetizing energy. According to Eq. (R2) and Fig. S2c, the change Δ/Δ_0 becomes weaker as H_K increases. In the limit $H_K \gg H_x$, the wall magnetization cannot be tilted, and the DW width is not affected by H_x . In contrast, applying H_y does not tilt the wall magnetization; instead, it aligns

¹⁴ Malozemoff, A. P., & Slonczewski, J. C. Magnetic domain walls in bubble materials: advances in materials and device research (Vol. 1). Academic press. chapter IV, 77-121 (1979). <https://doi.org/10.1016/B978-0-12-002951-8.50008-X>.

the wall magnetization to the field direction to lower the Zeeman energy. Consequently, the DW expands under H_y , see Eq. (R3).

Quantitatively, the maximum change in DW width under a 40-mT H_x is estimated to be only 0.8%, and becomes even smaller for larger H_K . This small variation may explain why the DW velocity remains almost constant under H_x in Fig. S2a. However, a H_y of 40 mT expands the DW width by up to 2.3%. As the reviewer pointed out, the increase in DW width can reduce the nonadiabaticity, accounting for the observed decrease in DW velocity under H_y in Fig. S2b. Previous theoretical studies demonstrated that nonadiabaticity inversely scales with DW width in terms of mistracking^{15,16,17}. Other potential mechanisms, such as in-plane-field-induced Walker breakdown, only affect the DW velocity in the presence of H_x , but not H_y . Specifically, H_x lowers the Walker breakdown threshold, while H_y suppresses it. In this case, the DW velocity should change under H_x but not under H_y . This is in contradiction to our experimental observations and consistent with DW motion remaining below the Walker limit. Additionally, we observed that DW velocity $v \gg u$, which further demonstrates the DW motion remains below the Walker limit, as $v = u$ would be expected once it enters the Walker regime. Thus, Walker-breakdown-induced velocity changes can be ruled out.

We have included a summary of this analysis in the main text and reported Fig. S2a-d and the full derivation of Eqs. R2 and R3 in the revised Supplementary Note 2.

¹⁵ Akosa, C. A. et al. Role of spin diffusion in current-induced domain wall motion for disordered ferromagnets. Phys. Rev. B 91, 094411 (2015). DOI: <https://doi.org/10.1103/PhysRevB.91.094411>.

¹⁶ Xiao, J., Zangwill, A. & Stiles, M. D. Spin-transfer torque for continuously variable magnetization. Phys. Rev. B 73, 054428 (2006). DOI: <https://doi.org/10.1103/PhysRevB.73.054428>.

¹⁷ P. Chureemart et al, Influence of uniaxial anisotropy on domain wall motion driven by spin torque. Phys. Rev. B 92, 054434 (2015). DOI: <https://doi.org/10.1103/PhysRevB.92.054434>.

Summary of changes

All changes to the text have been highlighted in blue in the revised manuscript and supplementary information files. Additionally, we have included the following changes:

Revised manuscript:

1. In lines #78-82 of Introduction Section and lines #247-253 of Discussion Section, we added discussions on the in-plane magnetic field dependence of the DW velocities.

Supplementary material:

1. In Supplementary Note 2, we added the full derivation of the influence of in-plane magnetic field on the DW width and nonadiabaticity, which explains the observed dependence of DW velocities on the in-plane magnetic fields.

Reviewer #2

(Remarks to the Author):

In my last report, I had no further remarks or questions and I recommended publication of the manuscript in Nature Communications. Having read the other reports, I note a disagreement between Referee 1 and the authors concerning the modeling. Referee 1 concludes that "... nothing is new if the authors rely on such a simplified model to explain the experimental results ...". On this point, I disagree and find the authors' response convincing. In particular, a new experimental finding does not necessarily require a new type of theory. Identifying a material with exceptional parameters within an established model can still represent a high degree of novelty. Overall, I continue to recommend publication.

Response: We sincerely appreciate the reviewer's positive assessment of the novelty of our work and the recommendation to publish this manuscript.

Reviewer #3

(Remarks to the Author):

Dear authors,

I am already convinced by the authors' responses in the previous round, and I recommended that the results be published in Nature Communications. In this round as well, other reviewers posed sharp questions, and I believe the authors provided convincing answers. Once again, I recommend publication.

Response: We thank the reviewer for positive remarks on our revised manuscript and the arguments presented in the rebuttal letter, and for recommending publication of the manuscript.

Reviewer #4

(Remarks to the Author):

First, the referee thanks the authors for their response regarding the additional experiments. The authors have experimentally verified the dependence of the domain wall (DW) velocity v on H_x and H_y as predicted by the 1D STT model. Nevertheless, even with this confirmation, and even if the authors argue that the application of the 1D model still allows them to extract important material parameters of NCO, I believe that significant limitations remain.

The authors emphasize that, based on the application of the 1D model, NCO exhibits a very large ratio $\beta/\alpha \sim 20$ and an exceptionally large nonadiabaticity $\beta \sim 1.4$. They claim that the observation of such a large β in NCO, compared to the previously studied GdFeCo ferrimagnet with $\beta \sim -0.5$, constitutes novelty. However, I believe that this interpretation potentially holds a fundamental issue arising from treating a two-sublattice ferrimagnetic system within a ferromagnetic 1D model.

According to a study on the damping parameter α in GdFeCo ferrimagnets (PRL 122, 127203 (2019)), the effective α in ferrimagnets can be significantly smaller than the value inferred from ferromagnetic FMR analysis. In the present work, the authors use $\alpha = 0.08$ for NCO. If this value is also subjected the same limitation—namely, interpreting a ferrimagnet as a ferromagnet, as implicitly done in the magnetization treatment within the 1D model—then the resulting analysis could change substantially.

Since the domain wall velocity scales with β/α , if the actual α is smaller than the assumed value, the resultant nonadiabaticity $\beta \sim 1.4$ may no longer be exceptionally large. Furthermore, while the authors claim $\beta/\alpha \sim 17-20$ for NCO, one the value is nearly 100 for GdFeCo from Ref. [Nat. Electron. 2, 389–393 (2019)].

Of course, domain wall motion at high velocity under low current density is an important result from an applications perspective. However, I find it difficult to agree with the authors' arguments that the application of the 1D model alone is sufficient to firmly establish the superior intrinsic properties of NCO—such as its claimed large nonadiabaticity and high spin polarization. For these reasons, I remain hesitant to recommend publication.

Response: We thank the reviewer for taking additional time to review our manuscript and for the positive remarks on our verification of the in-plane-field dependence of DW motion. Below, we further address the reviewer’s concerns with a point-by-point response.

First, we do not emphasize novelty by comparing the magnitudes of $\frac{\beta}{\alpha}$ and β between GdFeCo and NCO. Instead, the large nonadiabatic torques observed in both materials provide strong experimental evidence for the important role of nonadiabatic torque in fast DW dynamics, which was highlighted as one of the key findings of this work. Specifically, $\frac{\beta}{\alpha} \ll 1$ for the typical ferromagnets, whereas it can be exceptionally larger than 1 (e.g., $\frac{\beta}{\alpha} \approx 20$ for NCO and $\frac{\beta}{\alpha} \approx -100$ for GdFeCo) in antiferromagnetically coupled ferrimagnets due to spin mistracking. The resulting giant nonadiabatic torque leads to fast ferrimagnetic DW motion compared to that of ferromagnets. It provides the first insights into the nonadiabatic torque responsible for fast DW dynamics.

Second, regarding the reviewer’s concern on the validity of 1D model to NCO, we note that only in rare-earth transition-metal ferrimagnets such as GdFeCo^{1,2}, Gd_xCo_{1-x}^{3,4} and Co_{1-x}Tb_x⁵, which possess finite angular momentum compensation temperature T_A , does the 1D model need to be modified to interpret DW dynamics in the vicinity of the compensation points. Furthermore, we carefully reviewed the ref [PRL 122, 127203 (2019)] and found that it is still based on the 1D model. The theoretical model is described in the Supplementary Information of Ref⁶. Starting from the 1D model with considering the two sublattices of a ferrimagnet, the authors derived the steady-state DW velocity under the magnetic field, given by:

$$v = \frac{(M_1 - M_2)\Delta}{\alpha_1 S_1 + \alpha_2 S_2} H, \quad (\text{R1})$$

¹ Kim, KJ. *et al.* Fast domain wall motion in the vicinity of the angular momentum compensation temperature of ferrimagnets. *Nat Mater* **16**, 1187–1192 (2017). DOI: <https://doi.org/10.1038/nmat4990>.

² Okuno, T. *et al.* Spin-transfer torques for domain wall motion in antiferromagnetically coupled ferrimagnets. *Nat Electron* **2**, 389–393 (2019). DOI: <https://doi.org/10.1038/s41928-019-0303-5>.

³ Caretta, L. *et al.* Fast current-driven domain walls and small skyrmions in a compensated ferrimagnet. *Nature Nanotech* **13**, 1154–1160 (2018). DOI: <https://doi.org/10.1038/s41565-018-0255-3>.

⁴ Cai, K. *et al.* Ultrafast and energy-efficient spin-orbit torque switching in compensated ferrimagnets. *Nat Electron* **3**, 37–42 (2020). DOI: <https://doi.org/10.1038/s41928-019-0345-8>.

⁵ Siddiqui, S. *et al.*, Current-induced domain wall motion in a compensated ferrimagnet. *Phys. Rev. Lett.* **121**, 057701 (2018). DOI: <https://doi.org/10.1103/PhysRevLett.121.057701>.

⁶ Kim, KJ., *et al.* Fast domain wall motion in the vicinity of the angular momentum compensation temperature of ferrimagnets. *Nature Mater* **16**, 1187–1192 (2017). DOI: <https://doi.org/10.1038/nmat4990>.

where $\alpha_{1,2}$ and $s_{1,2}$ are the sublattice damping parameters and spin densities. If we adopt a general definition of Gilbert damping⁷, and consider two sublattices for a ferrimagnet^{8,9}, the effective damping is given by $\alpha_{\text{eff}} = \frac{\alpha_1 s_1 + \alpha_2 s_2}{s_1 - s_2}$, with a corresponding Rayleigh dissipation function $\mathcal{R} = \alpha_{\text{eff}} |s_1 - s_2| \int dV \dot{n}^2 / 2$ where n is the Néel vector. The DW velocity of Eq. R1 becomes:

$$v = \frac{(M_1 - M_2)\Delta}{\alpha_{\text{eff}}(s_1 - s_2)} H. \quad (\text{R2})$$

There are two cases for the Eq. (R2) depending on the temperature, as described below.

Case 1: The temperature is far away from the angular momentum compensation temperature T_A , where the net angular momentum density $L_S = s_1 - s_2 \neq 0$. Substituting $s_1 - s_2 = \frac{M_1 - M_2}{\gamma_{\text{eff}}}$, one gets the DW velocity:

$$v = \frac{\gamma_{\text{eff}} \Delta}{\alpha_{\text{eff}}} H, \quad (\text{R3})$$

which has the equivalent format as that for a ferromagnet¹⁰.

Case 2: The temperature approaches T_A where $L_S = s_1 - s_2 \rightarrow 0$. In this circumstance, the damping parameter α_{eff} diverges under the definition of $\alpha_{\text{eff}} = \frac{\alpha_1 s_1 + \alpha_2 s_2}{s_1 - s_2}$, which may lead to an overestimation of α_{eff} extracted from ferromagnetic FMR analysis^{8,9}. Therefore, some works have introduced the new definition of damping parameter $\alpha'_{\text{eff}} = \frac{\alpha_1 s_1 + \alpha_2 s_2}{s_1 + s_2}$ with the associated Rayleigh dissipation function $\mathcal{R}' = \alpha'_{\text{eff}} (s_1 + s_2) \int dV \dot{n}^2 / 2$ to avoid divergence of the damping parameter

⁷ Gilbert T L. A phenomenological theory of damping in ferromagnetic materials. *IEEE transactions on magnetics*, **40**, 3443-3449 (2004) DOI: [10.1109/TMAG.2004.836740](https://doi.org/10.1109/TMAG.2004.836740).

⁸ C. D. Stanciu *et al.* Ultrafast spin dynamics across compensation points in ferrimagnetic GdFeCo: The role of angular momentum compensation. *Phys. Rev. B* **73**, 220402(R) (2006). DOI: <https://doi.org/10.1103/PhysRevB.73.220402>.

⁹ M. Binder *et al.*, Magnetization dynamics of the ferrimagnet CoGd near the compensation of magnetization and angular momentum. *Phys. Rev. B* **74**, 134404 (2006). DOI: <https://doi.org/10.1103/PhysRevB.74.134404>.

¹⁰ Beach, G., *et al.* Dynamics of field-driven domain-wall propagation in ferromagnetic nanowires. *Nature Mater* **4**, 741–744 (2005). <https://doi.org/10.1038/nmat1477>.

at $T_A^{11,12}$. However, a later study argues that this is not necessary, as the Rayleigh dissipation rate is actually determined by $L_\alpha = \alpha_{\text{eff}}L_S = \alpha_{\text{eff}}(s_1 - s_2)$, which does not diverge at T_A^{13} .

Nevertheless, ferrimagnets without compensation point, such as NCO, Bi-YIG¹⁴, Mn₄N¹⁵, do not suffer from the above issue and the 1D model can be directly applied to describe their DW dynamics in the whole temperature range. Moreover, in the experiment, we directly extracted the damping parameter from the characteristic time of DW inertia, given by $\tau = \frac{1+\alpha^2}{\alpha\gamma H_K}$, instead of FMR. Both aspects ensure an appropriate extraction of the damping parameter without overestimating it.

Third, there is a difference in magnitudes of $\frac{\beta}{\alpha}$ between GdFeCo and NCO; however, this does not imply that either system is superior. The values remain within the same order of magnitude and are still orders of magnitude larger than those typically observed in ferromagnets. In addition, as stated in line # 228-231 of the main text, this difference may arise from either intrinsic material parameters or from the difference in experimental approaches.

Finally, as the reviewer noted, a key achievement of this work is to demonstrate that STT can drive fast DW motion under a small current density in the single-layer ferrimagnet NCO in the absence of magnetic field. Such exceptionally high performance was successfully explained by the 1D model in combination with large nonadiabaticity, small magnetization, and high spin polarization. Notably, the large nonadiabaticity originates from the antiferromagnetically coupled sublattices, while the high spin polarization arises from the unique band structure near the Fermi level of NCO. The 1D model helps us parametrize the origin of the exceptional STT performance in NCO, providing a useful phenomenological framework to optimize and compare materials enabling high DW mobility. From this perspective, we believe that the 1D model is sufficient for

¹¹ D. Kim *et al.*, Low Magnetic Damping of Ferrimagnetic GdFeCo Alloys. *Phys. Rev. Lett.* **122**, 127203 (2019). DOI: <https://doi.org/10.1103/PhysRevLett.122.127203>.

¹² Okuno, T., *et al.* Spin-transfer torques for domain wall motion in antiferromagnetically coupled ferrimagnets. *Nat Electron* **2**, 389–393 (2019). DOI: <https://doi.org/10.1038/s41928-019-0303-5>.

¹³ E. Haltz *et al.*, Domain wall dynamics in antiferromagnetically coupled double-lattice systems. *Phys. Rev. B* **103**, 014444 (2021). DOI: <https://doi.org/10.1103/PhysRevB.103.014444>.

¹⁴ Caretta, L. *et al.*, Relativistic kinematics of a magnetic soliton. *Science* **370**, 1438-1442 (2020). DOI: [10.1126/science.aba5555](https://doi.org/10.1126/science.aba5555).

¹⁵ Gushi, T. *et al.*, Large current driven domain wall mobility and gate tuning of coercivity in ferrimagnetic Mn₄N thin films. *Nano Letters* **19**, 8716-8723 (2019). DOI: [10.1021/acs.nanolett.9b03416](https://doi.org/10.1021/acs.nanolett.9b03416).

the scope of this work. Nevertheless, we note that the 1D model does not predict which material exhibits giant nonadiabatic STT, or high spin polarization.

Summary of changes

All changes to the text have been highlighted in blue in the revised manuscript and supplementary information files. Additionally, we have included the following changes:

Revised manuscript:

1. In line #613 of caption in Fig. 1a, we added: “The structure was generated using VESTA software”.